



# Effect of sea breeze circulation on aerosol mixing state and radiative properties in a desert setting

Yevgeny Derimian[1], Marie Choël[2], Yinon Rudich[3], Karine Deboudt[4], Oleg Dubovik[1], Alexander Laskin[5], Michel Legrand[1], Bahaiddin Damiri[1,6], Ilan Koren[3], Florin Unga[1,2], Myriam Moreau[2], Meinrat
O. Andreae[7] and Arnon Karnieli[8]

[1]Laboratoire d'Optique Atmosphérique, UMR8518 CNRS, Université de Lille1, Villeneuve d'Ascq, 59655, France
[2]Laboratoire de Spectrochimie Infrarouge et Raman, Université de Lille 1, Villeneuve d'Ascq, 59655, France
[3]Department of Earth and Planetary Sciences, Weizmann Institute of Science, Rehovot 76100, Israel
[4]Laboratoire de Physico-Chimie de l'Atmosphère, Université du Littoral Côte d'Opale, Dunkerque, 59140, France
[5]William R. Wiley Environmental Molecular Sciences Laboratory, Pacific Northwest National Laboratory, P.O. Box 999, MSIN K8-88, Richland, WA, 99352, USA
[6]Cimel Electronique, Paris, 75011, France
[7]Biogeochemistry Department, Max Planck Institute for Chemistry, P.O. Box 3060, 55020 Mainz, Germany
[8]Remote Sensing Laboratory, Jacob Blaustein Institutes for Desert Research, Ben-Gurion University of the Negev, Sede
Boker, 84990, Israel

*Correspondence to*: Yevgeny Derimian (Yevgeny.Derimian@univ-lille1.fr)

**Abstract.** Chemical composition, microphysical and optical properties of atmospheric aerosol deep inland in the Negev Desert of Israel were found be influenced by daily occurrences of sea breeze flow from the Mediterranean Sea. Abrupt
increases in aerosol volume concentration and shifts of size distributions towards larger sizes, which are associated with increase in wind speed and atmospheric water content, were systematically recorded during the summertime at a distance of at least 80 km from the coast. Chemical imaging of aerosol samples confirmed an increased contribution of highly hygroscopic particles during the intrusion of the sea breeze. Besides a significant fraction of marine aerosols, the amount of internally mixed marine and mineral dust particles was also increased during the sea breeze period. The number fraction of
marine and internally mixed particles during the sea breeze reached up to 88 % in the PM1-2.5 and up to 62 % in the PM2.5-10 size range. Additionally, numerous particles with residuals of liquid coating were observed by SEM/EDX analysis. Ca-rich dust particles that had reacted with anthropogenic nitrates were evidenced by Raman microspectroscopy. The resulting hygroscopic particles can deliquesce at very low relative humidity. Our observations suggest that aerosol hygroscopic growth in the Negev Desert is triggered by the daily sea breeze arrival. The varying aerosol microphysical and optical
characteristics perturb the solar and thermal infrared radiation. The changes in aerosol properties induced by the sea breeze, relative to the background situation, doubled the shortwave radiative cooling at the surface (from -10 to -20.5 Wm⁻²) and increased by almost three times the warming of the atmosphere (from 5 to 14 Wm⁻²), as evaluated for a case study. Given the important value of observed liquid coating of particles, we also examined the possible influence of the particle homogeneity assumption on the retrieval of aerosol microphysical characteristics. The tests suggest that sensitivity to the coating appears



if backward scattering and polarimetric measurements are available for the inversion algorithm. This may have an important implication for retrievals of aerosol microphysical properties in remote sensing applications.

## 1 Introduction

Chemical composition and mixing state of atmospheric particles evolve during their transport in the atmosphere leading to
changes in the aerosol optical properties and radiative effect. For instance, airborne mineral dust particles, which are often modeled as hydrophobic particles since they are originally composed of non-soluble chemical species, can be transformed into complex heterogeneous mixtures of non-reactive and reactive compounds (Dentener et al., 1996; Krueger et al., 2003; Falkovich et al., 2004; Krueger et al., 2004; Laskin et al., 2005b). The appearance of secondary organics on the dust surface (Falkovich et al., 2004) and heterogeneous reactions between pollutants and components of dust can produce a deliquescent
layer that favors water uptake by mineral dust (Usher et al., 2003; Laskin et al., 2005a). Also, laboratory experiments have shown that water adsorption can occur even on non-reactive surfaces of dust particles (Navea et al., 2010). Numerous field observations provide evidence of the presence of water-soluble inorganic ions such as sulfates and nitrates as dust coating material (Levin et al., 1996; Levin et al., 2005). Therefore, airborne mineral dust can be treated as a potential surface for heterogeneous chemistry (Usher et al., 2003), which hygroscopic, morphological, and optical properties are significantly
altered during its atmospheric lifetime. In the case of ambient aerosols, when the changes occur in time and space, identification and evaluation of the physico-chemical transformations and their influence on radiative properties can be particularly complex. In the current study, we show that a rather regular sea breeze phenomenon can be a test case for exploring how the mixing state of airborne particles changes under conditions of mixed air mass and varying relative humidity. During sea breeze intrusions inland, marine particles can mix with local pollutants in urban/industrial areas or with
aeolian dust in arid regions and heterogeneous reactions can take place. The interactions can produce more complex atmospheric particles, with microphysical and optical properties that may be difficult to model. In our study, we focus on transformations of aerosol optical and physico-chemical properties during sea breeze intrusions into the Negev Desert of Israel.

The Negev Desert is known be generally influenced by airborne mineral dust, urban/industrial pollution and marine aerosols
(Maenhaut et al., 1997; Andreae et al., 2002; Sobanska et al., 2003; Karnieli et al., 2009). However, the influence of the daily sea breeze on the aerosol properties in the Negev Desert was not studied so far. In this study, we explore the influence of penetrating marine air masses on the mixing state and hygroscopic properties of aerosol particles observed at Sede Boker, a desert site located 80 km away from the Mediterranean coast (the site sometimes also referred to as Sde Boker). The dust at this site originates from either local or remote dust sources. The pollutants originate in the central and more polluted areas of
Israel or are transported from Eastern Europe (Andreae et al., 2002; Karnieli et al., 2009; Maenhaut et al., 2014). While windblown desert dust affects the Negev Desert all year long with concentration peaking in spring and autumn, a maximum of the anthropogenic aerosols appears in the summer time (Derimian et al., 2006). Long-term observations at the site provide





an extensive dataset of aerosol characteristics and origins. The regular intrusion of the sea breeze is now employed for elucidation of how the evolution of humid conditions accompanied with the intrusion of aged sea-salt and pollution aerosols modifies the mixing state of mineral dust and how this affects the aerosol radiative effect. To address this goal, we employ a multidisciplinary approach by utilizing a combination of comprehensive remote sensing observations coupled with *in-situ*

aerosol measurements and *off-line* chemical imaging of atmospheric particles collected at the site. The comprehensive observations were conducted during summer of 2012. Following aerosol particle sampling, *off-line* scanning electron microscopy, X-ray microanalysis and Raman spectroscopy were employed for chemical imaging of particles. Finally, effects of internally mixed particles on their optical properties and consequent implications for remote sensing algorithms are also discussed.

**2 Measurement site and meteorological conditions**

The Sede Boker site is located in the Negev Desert, in the southern part of Israel (30°51'N, 34°47'E) and is remote from big cities and industrial areas. It is about 80 km inland from the Mediterranean Sea coast and 470 m above sea level. As a remote desert site for atmospheric aerosol observations, the Sede Boker site was established in 1995 as part of the AERONET network of sun/sky photometers and the ARACHNE program, e.g., (Ichoku et al., 1999; Formenti et al., 2001; Andreae et

al., 2002).

The main aerosol types that affect the site are local and long-range transported mineral dust, transported pollution and marine particles. The air masses in the summer period originate from the north-west, bringing anthropogenic aerosols from densely populated area of central Israel and from Eastern Europe (Andreae et al., 2002). The transport of anthropogenic emissions is attributed to a persistent large-scale synoptic condition that is characterized by a semi-permanent low-pressure

trough extending from the Persian Gulf to the Negev. Diurnal variations of mixed layer depth in this time are driven by surface heat fluxes and by the daily sea breeze cycle (Dayan and Rodnizki, 1999). During the observation period in summer 2012, measurements from a local meteorological station show generally northwesterly wind direction with a regular sea breeze presence in afternoon. Figure 1a shows that after about 14:00 UTC (the local time is UTC+3) the mean wind direction is relatively constant and the mean wind speed increases up to 8 m s⁻¹. Although, the average air temperature and relative

humidity near the ground (Fig. 1c) show a smooth behavior, abrupt changes can be clearly distinguished in observations for any single day when the sea breeze occurs. An example of a clear manifestation of the sea breeze arrival at 14:00 UTC is presented for August 16 (Figures 1b, d), i.e., drop of temperature, rise of relative humidity, increase of wind speed accompanied by stabilized wind direction from north-west. August 16 is the case study day when aerosol sampling before and during the sea breeze was analyzed. This specific day is selected for a comprehensive analysis and in-depth

understanding of aerosol properties in the Negev Desert during the sea breeze phenomenon.

In addition to the local meteorological measurements, 24-hour and 3-day backward trajectories are obtained using the HYSPLIT model for August 16 (4). The 3-day trajectories show a general north-west air mass origin (Fig. 2c) that is typical





for the summer season and the 24-hour trajectories show a change in direction of the near-ground air masses (red line for 10 m altitude) when the sea breeze arrives to the site (Fig. 2a, b). Prior to arrival at the measurement site, the air masses remain most of the time over the Mediterranean Sea (Fig. 2c), then, penetrate into the land over the densely populated Gaza Strip, and advance over the desert area for several hours. The backward trajectories presented in Fig. 2 correspond to the time

when the samplings were started before and during the sea breeze, i.e. 13:00 and 14:30 UTC, respectively. The model also shows that the air masses and therefore the transported aerosol particles at altitudes of 10 m and 500 m are exposed to 60 to 80 % RH levels several hours before sampling (see bottom panels in Fig. 2). The model also points out some increase in RH (from 32 to 36 %) at 10 m above ground level (AGL) when the sea breeze starts. Although, the RH values provided by HYSPLIT are somewhat different from the corresponding values measured by the local meteorological station (increases

from 26 to 43 %), the increasing tendency is consistent.

## 3 Instrumentation

### 3.1 Remote sensing setup

#### 3.1.1 Sun/sky photometer

The sun/sky photometric measurements at the Sede Boker site are performed as part of the global Aerosol Robotic Network

(AERONET) (Holben et al., 1998). The measurements are conducted with a photometer manufactured by CIMEL Electronique, Paris, France. The automatic direct sun photometric measurements are normally conducted every 15 minutes and provide spectral aerosol optical thickness at 340, 380, 440, 500, 675, 870, and 1020 nm nominal wavelengths. The 940 nm channel is used to retrieve the atmospheric water vapor content. The angular distribution of sky radiance is measured at 440, 670, 870, and 1020 nm. The measured spectral sun and sky radiances are used for retrieval of aerosol optical parameters

at four wavelengths by the AERONET inversion code (Dubovik and King, 2000; Dubovik et al., 2006) that employs models of homogeneous spheres and randomly oriented spheroids. The spectral aerosol optical thickness measurements are also used for calculating the Ångström exponent (Å) that is an indicator of aerosol size. For instance, between the wavelength of 440 nm and 870 nm, Å  is calculated as

$$\text{Å} = -\frac{ln\left(\frac{\tau_{870}}{\tau_{440}}\right)}{ln\left(\frac{\lambda_{870}}{\lambda_{440}}\right)},\tag{1}$$

where $\tau$ is the aerosol optical thickness (AOT) and $\lambda$ is the wavelength. The Ångström exponent below 0.5 indicates an important contribution of coarse mode aerosols, the range between 0.5 and 1.0 corresponds to a bimodal size distribution, and a value above 1.0 indicates a dominant fine mode aerosol.



### 3.1.2 Thermal infrared radiometer

The multichannel thermal infrared (TIR) radiometer is designed to measure thermal radiation emitted by the atmosphere and surface system. The instrument has been developed in collaboration between the Laboratory of Atmospheric Optics (LOA) of University of Lille (Legrand et al., 2000; Brogniez et al., 2003) and CIMEL Electronique. This is the same manufacturer

as that of the AERONET photometers and, therefore, both instruments have convenient similarities in protocol of functionality that facilitates in-field operation. The TIR radiometer provides radiances and brightness temperature of a target viewed in about 10 degree full field of view. The instrument employed at the Sede Boker site was operated at three 1-µm narrowband spectral channels centered at 8.6, 10.8, and 12.0 µm and at an extra broadband channel covering the spectrum from 8 to 14 µm. The instrument operates in a sky-scanning mode and in this work, the analyzed values are the sky

brightness temperature from vertical upward looking position. The radiometer is equipped with a humidity sensor in order to shut down automatically in case of precipitation or dew to prevent water deposition on the detector. In addition, the system can shut down the instrument when relative humidity is about 80 %, which limits the number of observations, mainly during nighttime. The instrument was set up at the site by LOA for a six-month experimental period and with the purpose of complementary and intensive observations.

### 3.1.3 LIDAR

The ground-based LIDAR observations at the Sede Boker site are conducted as part of the NASA Micro-Pulse Lidar Network (MPLNET) (Welton et al., 2001), wherein sites are generally co-located with the AERONET sites. The MPLNET is a federated network of micropulse LIDAR systems (Spinhirne et al., 1995; Spinhirne et al., 2002) that uses standardized calibrations, operational protocols and processing. The network is supported by NASA Earth Observing System program

(Wielicki et al., 1995). Data products at three levels of processing provide real-time normalized relative backscatter, aerosol and cloud heights, and optical property retrievals (Campbell et al., 2002; Welton and Campbell, 2002), http://kimura.gsfc.nasa.gov/data. In our study, we employ only the vertical distribution of LIDAR backscatter signal at 532 nm for the purpose of illustration of vertical and temporal variability of the aerosol loading.

### 3.1.4 Broadband solar flux

The Solar Radiation Network (SolRad-Net, http://solrad-net.gsfc.nasa.gov) is associated with the AERONET network of federated ground-based sensors that provides high-frequency solar flux measurements in quasi-real time. Similar to MPLNET, the sites are co-located with AERONET, and standardized calibrations and operational protocols are applied to the measurements. In general, SolRad-Net provides measurements from several flux instruments including filtered and unfiltered pyranometers, photosynthetically active radiation (PAR) and ultraviolet (UV-A and UV-B). In this study we use

the broadband shortwave solar spectrum (0.3 – 2.8 µm) irradiance, measured by a Kipp and Zonen CM-21 pyranometer. The





data correspond to the quality level 1.5, which have been cloud screened, cleared of any operational problems and pre-calibrated. The instantaneous irradiance analyzed at the Sede Boker site is recorded at 10-minute intervals.

### 3.2 Backward Trajectories

The air mass backward trajectories are obtained using the 3-D HYSPLIT (HYbrid Single-Particle Lagrangian Integrated Trajectory) model of the U. S. National Oceanic and Atmospheric Administration (NOAA) (Draxler and Hess, 1998). The runs for backward trajectories are performed using the global data assimilation system. It is performed for altitudes of 10 m, as an indicator of near the surface air mass origin, and 500 m and 1000 m above ground level. The relative humidity at the corresponding altitudes and time are also provided by HYSPLIT.

### 3.3 In-situ measurement and sampling

#### 3.3.1 Integrating nephelometer

Near ground, the light scattering extinction coefficient at 545 nm is measured with 2 min temporal resolution by a single-wavelength integrating nephelometer (M903, Radiance Research, Seattle, WA, USA). The inlet is located outdoors on a roof at about 10 m above ground and faces downward. The instrument itself is situated indoors and air is supplied through plastic tubing of up to 3 m length and of 2.2 cm internal diameter. The instrument was set up in November 1999 and was regularly

calibrated in the field until November 2003. Variability of the calibration coefficients during this time was within 6%. A different strategy was applied afterwards when a series of reference tests with particle-free air and $CO_2$ as a calibration gas were periodically conducted; the procedure enables to trace the variability of the calibration coefficients and apply correction to the measured values. In this study we do not intend to evaluate the long-term temporal trend; the observations are used only for confirmation of the response of near-ground aerosol optical properties to the sea breeze arrival. Thus, the

abovementioned corrections are not needed in the analysis of diurnal variability of the scattering coefficient on a specific day. Relative humidity (RH) in the scattering volume of the instrument is also of importance since a nonlinear increase in the scattering coefficient is possible when RH is above 80% (Andreae et al., 2002). Thus, early morning and late evening data, when the RH is elevated, should be interpreted as a high limit. Nevertheless, the behavior of the measured scattering coefficient observed in this study is generally consistent with other independent measurements at the site.

#### 3.3.2 Aerosol sampling

Aerosol samples were collected on the rooftop terrace of a three-story building, adjacent to the nephelometer inlet. Ambient particles were sampled before and during sea breeze flow using a three-stage cascade impactor (PM-10, Dekati Ltd.) at a flow rate of 10 L min-1. The nominal cut-off sizes (i.e., aerodynamic diameters at 50% of collection efficiency) of the impactor stages were: 10, 2.5 and 1 μm, respectively. Sampling durations ranged from 15 min to 1 h, depending on the

ambient aerosol load. Particles were impacted simultaneously onto 200-mesh copper TEM grids with carbon type-B





supporting films (Ted Pellar, Inc.) and Nuclepore$^{TM}$ polycarbonate membranes for SEM/EDX particle microanalysis. Additionally, particles were collected on glass slides for Raman micro-spectrometry. Samples were sealed in aluminum foil bags and stored at 4°C pending analysis. Among several samples collected during the campaign, samples from August 16, 2012 were selected as representative of the described phenomenon and are presented here in detail. The sampling time and duration were most successful for representing the before and during the sea breeze conditions, the phenomenon itself was also well pronounced and measured by all other instruments. The sampling conditions for this day are reported in Table 1 and discussed in detail in Sect. 2.

### 3.4 Chemical characterization at the particle scale

Offline-laboratory chemical imaging of the sampled atmospheric particles was carried out using SEM/EDX and Raman microspectroscopy.

### 3.4.1 Scanning electron microscopy with energy-dispersive X-ray spectrometry (SEM/EDX)

Single particle analysis by SEM/EDX was performed with a FEI Quanta 200 SEM equipped with an ultrathin-window energy-dispersive X-ray detector enabling the analysis of elements with atomic number higher than boron (Z≥5). However, for samples collected on polycarbonate membranes, elements lighter than sodium (Z<11) were not quantified because of high absorption within the samples due to carbon coating and substrate material. Automated particle analysis was run using the commercially available Link ISIS Series 300 Microanalysis system (Oxford Instruments®). The procedure of automatic particle recognition and analysis is described elsewhere (Choel et al., 2005). X-ray spectra were acquired with an acquisition time of 30 s, at an accelerating voltage of 20 kV and a probe current adjusted to 200 pA. The identification of individual particles is based on their elemental composition obtained from SEM/EDX data: the procedure and the criteria were described in a previous study (Deboudt et al., 2010). Elemental composition of particles is reported in this work as normalized atomic percent. Figure 3 shows the particle-classification chart used in the case of Negev particles. To elucidate the mixing state of particles, the analyzed particles were sorted into four different groups: Dust, Marine, Mixed Dust/Marine, and other. Particles sorted into the 'Dust' particle type were composed of silicate (Si-rich), aluminosilicate (Al- and Si-rich), calcite (Ca-rich), dolomite (Ca- and Mg-rich), gypsum (Ca- and S-rich), and Ti-rich particles. The 'Marine' particle type comprises fresh (Na- and Cl-rich) and aged (Cl-depleted) sea-salts. Particles that contain sea-salts internally mixed with crustal elements were assigned to the 'Mixed Dust/Marine' particle type. Particles not assigned to the previous particle types were sorted into the 'other' particle type comprising notably Mg-, S-, K-, and KCl-rich particles. Complementary manual examination of particles was performed using a HORIBA S-4700 field emission scanning electron microscope (FEG-SEM).

### 3.4.2 Raman microspectroscopy

Raman spectra were recorded for atmospheric particles in the coarse fraction (i.e., PM2.5-10). The glass plates with impacted particles were directly mounted on the microscope stage of a LabRAM HR confocal Raman microscope (Horiba



Scientific) equipped with an Olympus 100× objective with a numerical aperture of 0.90. Raman scattering was excited at 632.8 nm using a He–Ne laser. The laser spot size focused on the sample was 0.9 μm. To avoid laser damage to the sample, a neutral density filter with an optical density value of 0.6 was used. Raman measurements were carried out at ambient conditions (~60% RH and 295 K). Raman spectral mapping provides the spatial distribution of the various molecular species

within heterogeneous samples. The acquisition of computer-controlled Raman maps consisted in recording spectra in a point-by-point XY scanning mode with a 1 μm step and 10 s of integration time. According to the diffraction grating of 300 grooves per mm used in this work, Raman spectra were acquired in the range 170–2440 cm$^{-1}$ with a spectral resolution of about 4 cm$^{-1}$. The data processing of Raman maps is the following. The baseline was estimated individually for each spectrum using Asymmetric Least Squares (ALS) proposed by (Eilers, 2003; Eilers and Boelens, 2005). The order of

differences d was set to 3 (classical value), whereas the trade-off parameter (lambda) and the asymmetry parameter (p) were optimized for each map by visual inspection of the estimated baseline and the spectra after correction. Best results were obtained for a p value of 0.01 and 10$^{8}$ for lambda. The color map was obtained from the baseline corrected data using the net Raman intensity signal at a specific wavelength over all point spectra. The difference in the center positions of characteristic Raman bands were selected in order to minimize overlap of the characteristic Raman peaks of the several compounds present

in the aerosol samples. For the color map, intensity close to zero corresponds to black and the maximum signal intensity to bright color.

**4 Remote sensing observations**

Similarly to the meteorological parameters described in Sect. 2, recurrent abrupt changes in atmospheric aerosol optical characteristics can be observed nearly every day during the summer time, which is true not only for 2012, but for all

preceding and subsequent years. An example of several consecutive days of atmospheric remote sensing measurements during August 2012 is presented in Fig. 4. It shows daily variability of AOT at 440 nm, Ångström exponent between 870 nm and 440 nm, total column water vapor, and sky brightness temperature from three channels of the thermal infrared radiometer. In conjunction with the meteorological parameters, an analysis of the data in Fig. 4 suggests that the abrupt changes in the remote sensing measurements coincide with sharp changes in the air-mass as the sea breeze arrives. The AOT

increases and the Ångström exponent decreases significantly when the sea breeze arrives (Fig. 4a). Decrease of the Ångström exponent indicates an increased contribution of large aerosol particles. Attention can also be drawn to the recurrent increase of the sky brightness temperature and change of its spectral dependence, as measured by the thermal infrared radiometer from the ground. It shows that the spectral radiative properties in the thermal infrared spectra change significantly during the penetration of the sea breeze (Fig. 4c), similar as in the solar spectra (Fig. 4a). While the variability

of AOT and the Ångström exponent in the solar spectrum is due to the change in aerosol particles properties only, several processes can cause variability of the sky thermal infrared emission. Atmospheric water vapor can absorb solar and thermal radiation, and emit thermal radiation; water droplets and aerosol particles can absorb and scatter solar and thermal radiation,





and emit thermal radiation. Thus, in general, the sky brightness temperature can increase either due to higher atmospheric water vapor content or due to appearance of large mineral dust particles or water droplets. Aerosol particles and droplets have primary radiative effects in the 10.8 μm channel of the radiometer, at the center of the 10 μm window where the atmospheric gaseous transmittance is maximum (and the sky brightness temperature minimum). Absorption of the thermal

radiation by water vapor is stronger in the channels centered at 8.6 mm and 12 mm, which are located near the edges of this window. The spectral dependence (represented by the ratio of brightness temperatures) between the atmospheric window channel of 10.8 mm and the channel of 8.6 mm, which is more affected by water vapor, indicates a stronger increase of the sky thermal emission at 10.8 mm relative to 8.6 mm during periods with sea breeze (Fig. 4c). Therefore, an increase of the brightness temperature ratio (10.8 mm to 8.6 mm) suggests the appearance of not only water in the gas phase, but also of

large particles or water droplets. Note also that the ratio of brightness temperatures in Fig. 4c is approaching the value of one at the time of sea breeze arrival, that is, the spectral dependence of brightness temperature is approaching to neutral, which is a typical characteristic of clouds. The brightness temperature can also increase due to the arrival of a warmer air mass. However, as the meteorological data shows, the arrival of the sea breeze is associated with cooler air, while the water vapor content and amount of aerosol increases (Fig. 4a, b).

A more quantitative interpretation of the TIR signal requires accurate radiative transfer computations that also require information about vertical profiles of the aerosol extinction, concentrations of gas phase species and temperature, which are not available for the site of interest. The same information is needed for evaluation of the aerosol radiative forcing in TIR. However, the presented TIR radiometer measurements and the diurnal behavior of the sky brightness temperature are already informative. It shows that increase in the amount of water vapor and large size aerosol is likely to increase the TIR radiative

warming at the surface that generally counteract the aerosol cooling effect in the solar spectrum. The TIR measurements are also in line with the photometric observations in the solar spectrum by AERONET. That is, an abrupt increase in the AOT is observed when the sea breeze arrives. Notably, the increase in the AOT is often screened as a cloud because normally the aerosol properties do not change so fast in time and the screening algorithm fails. An example of the unscreened data (level 1.0) and the data after the cloud-screening algorithm is applied (level 2.0) is presented in Fig. 4a (see especially August 14).

In order to understand and describe in details the phenomenon, we focus on a specific, but typical, day (August 16) when a sampling of aerosols was conducted and analyzed in conjunction with remote sensing observations before and during the sea breeze. Figure 5 shows that at around 14:00 UTC (17:00 local time) the AOT, total column water vapor, scattering coefficient at the ground level, and sky brightness temperature have a sharp increase, while the Ångström exponent decreases. The LIDAR backscatter signal also increases in altitudes up to 1.8 km. The phenomenon reaches a maximum at

the sea breeze front and then decays gradually. The photometer acquisition is stopped after 16:00 UTC because of the low sun, but the LIDAR measurement, the thermal infrared radiometer, and the nephelometer remain available. From the temporal variability of the signal of these three instruments one can estimate that the effect of the sea breeze lasted until about 17:00 UTC, i.e., for about three hours. It can also be noted that some gradual increase in water vapor and brightness temperature starts already about two hours before the front of the sea breeze arrives. The increase of the water vapor towards





noontime is a usual process related to increasing temperature and evapotranspiration that influences the thermal infrared signal. It is noteworthy that the increase in water vapor is also correlated with a gradual decrease of the Ångström exponent, i.e., increase of the aerosol size, which can also be responsible for a gradual increase of the sky brightness temperature before the abrupt change occurs. In addition to the column-integrated remote sensing measurements, the scattering

coefficient, which is measured by nephelometer near the ground, shows that the abrupt change in aerosol characteristics occur also at the surface level (Fig. 5c). The fact supports representativeness of the particle sampling described in Sect 3.3.2. The diurnal variability of scattering coefficient is also correlated with the ambient RH (Fig. 1d). This is despite the nephelometer generally dry the aerosols inside the measurement volume and the RH inside the instrument is much more stable than the ambient RH. The variability of the scattering coefficient can be due to either change in aerosol concentration

or microphysical characteristics, like size, but it is difficult to draw a conclusion based on only a single wavelength measurement. In summary, all the abovementioned observations of the aerosol optical properties in the solar spectrum and radiation in the thermal infrared wavelength region manifest a coherent abrupt response to the increasing content of atmospheric water, associated with the sea breeze.

In order to examine the changes in the aerosol microphysical parameters that take place during the sea breeze, we use the

remote sensing observations of the aerosol volume size distribution and the complex refractive index as retrieved by the AERONET algorithm. On August 16, the average volume size distribution during the sea breeze is significantly different from the size distribution before the sea breeze (the averages are calculated for three observations during the sea breeze and five observations prior the sea breeze arrival). It shows an increase of the volume concentration and a size shift towards large sizes (Fig. 6a, b). The aerosol volume concentration is a product of particle number concentration and volume of given

particles. Thus, both the number concentration and the particle size may contribute. Indeed, stronger wind speed during the sea breeze can lift aerosols along the path of transport and increase the aerosol number concentration. However, as Fig. 6b shows, the radii of the particles are also increasing. The volume size distribution in Fig. 6b is normalized to the total volume concentration in order to enable a better comparison of the distribution shapes, which emphasizes the shift of the radii. It can be noted that the average water vapor concentration is also increasing from 1.4 to 2.2 g cm$^{-2}$. Figures 5c and d present

average size distributions obtained for 51 days when the AERONET inversions are available and the sea breeze is clearly observed in the meteorological data during summer 2012. Variability of the water vapor concentration is generally important on such days, and for convenience, the averages are calculated for three different ranges of the water vapor concentration. It shows that a shift in size distribution, similar to on August 16, occurs also in the three months' data of summer 2012. It appears also that the particles of the fine mode are affected much more strongly than those of the coarse mode. We then do a

similar analysis for March and April of the same year (24 days are analyzed) when the aerosol regime in the Negev Desert is very different and is governed mainly by African dust transport. The average water vapor during this dry air-mass transport does not exceed 2 g cm$^{-2}$ and no shift is observed either in the fine or in the coarse modes of the size distributions (Fig. 6e, f). Also the maximum of the coarse mode during the spring is at about 2 mm in contrast to 2.5 – 3 mm during the summer.



The real and imaginary parts of the complex refractive index and their spectral dependences are related to the aerosol particles' chemical composition. As the real part of the refractive index of water in visible spectrum is 1.33, it is expected that the real part of the refractive index of water-containing aerosols will decrease and approach the value of water. Figure 7 shows that this is the case for observations during the sea breeze (Fig. 7a) and for sea breeze days associated with increased

water vapor concentration (Fig. 7c). Curiously a decrease of mean real refractive index with increasing water vapor is also notable for the dust case of spring 2012 (Fig. 7e). However, the standard deviations are very high in this case (presented as error bars in Fig. 7) and there is no significant change in the corresponding size distributions (Fig. 6f). The imaginary part of the complex refractive index of pure water in the visible spectrum is practically zero; the marine aerosol, for example, is known to be non-absorbing (Dubovik et al., 2002). It is therefore expected that the imaginary part will also decrease with

increasing water content in the aerosol. The observations show, however, that the imaginary part for all analyzed cases slightly increases (Fig. 7b, d, f). Indeed, since the sea breeze air-masses can bring pollution aerosols, it is suggested that the reason for the increase of the imaginary part may be the presence of absorbing carbonaceous particles. The imaginary part also increases for the dust case, where we do not expect a carbonaceous aerosol contribution. At the same time, the standard deviations are particularly large for the imaginary part and it has to be mentioned that sensitivity of the AERONET

measurements to the complex refractive index is rather limited, which complicates any solid conclusion; sensitivity to the changes in aerosol size distribution, however, is quite high (Dubovik et al., 2000) since the sun photometer measures primary the forward scattered radiation, which strongly depends on the particle size.

The aerosol single scattering albedo (SSA), which is defined as the ratio of the scattering coefficient to the total extinction coefficient and represents the scattering effectiveness in total extinction, is one of the key parameters that determines the

aerosol radiative effect. With increasing imaginary part of the refractive index, the SSA becomes generally lower, indicating a stronger contribution of aerosol absorption. However, the SSA depends also on the aerosol size or more precisely on the size parameter, which is defined as the ratio of the particle size to the wavelength of light. The decrease of the SSA on August 16 was quite significant at all the corresponding wavelengths (440/670/8701020 nm); that is, a decrease from 0.97/0.96/0.96/0.96 before the sea breeze to 0.93/0.92/0.91/0.91 during the sea breeze, respectively.

## 5 Individual particle analysis

### 5.1 Elemental analysis of particles by SEM/EDX

Additional insights on the microphysical properties and mixing state of ambient particles are provided by computer-controlled SEM/EDX (CCSEM/EDX) analyses of aerosols sampled before and during the sea breeze on August 16, 2012. A total of 2077 particles were analyzed. Each particle was assigned to one of the particle types, defined in Sect. 3.4.1: Marine,

Dust, Mixed Dust/Marine, and Other. The relative proportions of these particle types are shown in Fig. 8.

The pie charts in Fig. 8 present particle-type fractions of particles in fine (PM1-2.5) and coarse (PM2.5-10) modes collected before and during the sea breeze event. Considering all analyzed particles for this sampling day (n = 2077), the most





abundant elements (excluding C, N, and O) were identified and the normalized average composition was calculated as $Na_1Mg_{0.08}Al_{0.11}Si_{0.31}S_{0.07}Cl_{0.04}K_{0.03}Ca_{0.28}Fe_{0.04}$. Overall, the 'Marine' particle type represented 48.4 % of all analyzed particles. Its average composition is $Na_1Mg_{0.04}S_{0.03}Cl_{0.04}$, indicative of nearly complete processing of sea-salt particles by $HNO_3$ and formation of $NaNO_3$ as a reaction product. The 'Dust' particle type accounted for 34.3 % of all analyzed particles.

Its average composition is $Mg_{0.05}Al_{0.36}Si1Ca_{0.82}Fe_{0.11}$, suggestive of aluminosilicates and calcium carbonates. The 'Mixed Dust/Marine' type contributed 14.6 % of all analyzed particles with an average composition of $Na_1Mg_{0.11}Al_{0.08}Si_{0.21}S_{0.1}Cl_{0.04}K_{0.05}Ca_{0.36}Fe_{0.04}$, which is typical of internal mixtures of dust and processed sea-salts. With an average composition of $Mg_{0.65}S_{0.65}Cl_{0.3}K_1$, the 'Other' particle group represented 2.7 % of all analyzed particles and comprised Mg-, S- and K-rich particles.

Before the sea breeze, dust was the predominant particle type in the coarse fraction (56 %) and was the second largest category in the fine fractions (35 %). An increase of both marine and mixed dust/marine particle types was clearly observed during the sea breeze. Therefore, the transport of marine particles during the sea breeze was confirmed by an increase from 37 to 44 % in the coarse fraction and from 55 to 61% in the fine fraction. Additionally, the internal mixing of dust/marine particles increased from 5 to 18 % in the coarse fraction and from 8 to 27 % in the fine fraction. It is noteworthy that during

the sea breeze a new type of particle was additionally detected, which contributed 2 and 7% of the analyzed particles in the fine and coarse fractions, respectively. These particles sorted into the 'Other' particle group were smaller than 1 µm in diameter and composed of potassium salts. Submicrometer-sized K-rich and KCl-rich particles can originate from biomass and waste burning emissions (Li et al., 2003), which however are not typical as known aerosol sources in the Negev Desert. However, given that the air masses passed over the densely populated coastal area (see trajectories over Gaza area in Fig. 2),

an anthropogenic source of these K-rich particles, e.g., from waste burning fires and cooking, is plausible. Also, the K-rich particles in the fine fraction have been already reported for the Sede Boker site previously (Formenti et al., 2001).

### 5.2 Number size distribution of particles by SEM/EDX

Figure 9 shows the particle number size distributions derived from analysis of images acquired by CCSEM/EDX. The presented radius is of a circle area equivalent, which represents the geometric properties. Note also that the maximal nominal

cut-off aerodynamic diameter of the analyzed stage of the impactor is 10 µm. Therefore, it should be realized that the size distributions in Fig. 9 and those retrieved from remote sensing in Fig. 6 are directly incomparable, for example, see discussions in (Reid et al., 2003). However, the size distributions per particles type and their relative variability can be informative. Thus, Fig. 9 shows total number size distributions and number size distributions for the four aerosol types separately before and during the sea breeze.

The total size distribution before the arrival of the sea breeze is mainly defined by dust and marine particles with concentration maxima at radii of about 0.25 µm and 0.75 µm, respectively (Fig. 9a). During the sea breeze, the total number size distribution is split into two modes with maxima centered at radii of about 0.4 µm and 1.75 µm (Fig. 9b). This is mainly due to contributions from possibly two types of marine particles – fresh and aged. The mean size of coarse marine particles





observed during the sea breeze is shifted towards larger sizes, i.e., an increase of radius from about 0.75 μm to 1.75 μm. This can be due to hygroscopic growth of more aged and larger particles exposed to the higher RH during the sea breeze. The number size distributions of dust and mixed dust/marine particle types are also changing during the sea breeze, with both size distributions broadening. The fact of the sea breeze influence is evident, but exact explanations can be complex. For

instance, a stronger contribution of coarse dust particles can also appear due to local aeolian resuspension of dust caused by the increased wind speed. The stronger contribution of the fine dust/marine particles is most probably due to freshly mobilized particles. Figure 9c shows the same total number size distributions as in Fig. 9a, b, but normalized to the total number of particles. This presentation facilitates a proper comparison of the distribution shapes and clearly illustrates the shift toward larger particle sizes during sea breeze, which is in line with the results obtained by the remote sensing.

However, the actual size shift may be even stronger because the SEM analysis provides partially dried aerosol size distributions. In addition, given the relative proportions of particle types, the fraction of hygroscopic particles can be estimated by the cumulative fractions of marine and mixed dust/marine particle types. This cumulative fraction significantly increases from 63% to 88% in the fine fraction and from 43% to 62% in the coarse fraction (Fig. 8), which supports the shift toward larger sizes during the sea breeze. Furthermore, the percentage of hygroscopic particles is largely underestimated by

an addition of 'Marine' and 'Mixed Dust/Marine' particle counts, if the fraction of hygroscopic dust was not taken into account. Previous studies reported that mineral dust in the Negev Desert predominantly consists of aluminosilicates and also calcium carbonates (Maenhaut et al., 1999). The solid calcium carbonate-containing particles can undergo heterogeneous reaction with gaseous nitric acid to form highly hygroscopic calcium nitrate particles. In fact, transformation of non-hygroscopic mineral dust into water soluble dust has been previously observed in aerosol samples collected in the Negev

Desert (Laskin et al., 2005a). In this study, we now subclassify all particles sorted in the 'Dust' particle type into five categories: aluminosilicates AlSi, Ca-rich, mix AlSi/Ca-rich, gypsum, and other AlSi. With an average composition of $Mg_{0.03}Al_{0.38}Si_1Fe_{0.09}$, the predominant subtype was AlSi accounting for 43.5% of dust particles, followed by mixed AlSi/Ca-rich particles representing 34.1% with an average composition of $Mg_{0.05}Al_{0.16}Si_{0.53}Ca_1Fe_{0.08}$. Ca-rich particles represented 17.4% of dust particles with an average composition of $Mg_{0.03}Ca_1$ typical of calcite and dolomite. With a frequency of 2.8%

and 2.2% respectively, gypsum particles ($S_{0.75}Ca_1$) and 'other AlSi' ($Al_{0.06}Si_{0.19}P_{0.23}S_{0.09}Ca_{0.78}Ti_1Fe_{0.4}$) were minor subtypes of dust particles, the latter comprising calcium phosphates and TiOx-rich aluminosilicates. To sum up, among particles sorted in the 'Dust' particle type, Ca-rich and mixed AlSi/Ca-rich particles accounted for 51.5% and could certainly be considered as hygroscopic dust. Further manual examination of the particles was performed to elucidate the nature of hygroscopic particles.

**5.3 Core-shell particle morphologies observed by SEM/EDX**

Analysis of the SEM/EDX observations also showed a large number of particles surrounded by halos (see particles marked by an arrow in Fig. 10a, b). Volatile components and water are lost due to the high-vacuum operating conditions in the SEM chamber and/or during metal coating. As a result, the dry residual compounds form halos around solid cores. This gives





direct evidence that the halos consist of residues of a hygroscopic surface layer after dehydration. In our sample, the halos were found on aged deliquescent marine (Fig. 10d), internally-mixed dust/marine (Fig. 10f) and dust particles. The presence of halos surrounding some dust particles confirms that the surface of the dust can be covered by potentially hydrophilic layers. The size of such coated dust particles may vary by hygroscopic growth during sea breeze events.

Figure 11 shows elemental maps and EDX spectra of an individual dust (AlSi/Ca-rich) particle with a halo. Calcium is relatively more abundant in the halo than in the core, pointing to a probable presence of liquid nitrate coating of dust in the form of calcium nitrate. As particles were collected on polycarbonate membranes, the detection of nitrogen is hampered. To confirm the presence of water-solvated nitrate coatings on some dust particles, complementary analysis was carried out using Raman micro-spectrometry.

**5.4 Raman maps of particles**

Complementary to elemental analysis by SEM/EDX, Raman micro-spectroscopy distinguishes between solid, deliquescent and solid inorganic nitrate compounds based on the nitrate band shift. In the liquid state, however, the characteristic nitrate band is identical to those of sodium and calcium nitrate. An example of the Raman molecular mappings is presented in Fig. 12. The spectral map of the 1086 cm$^{-1}$ peak, attributed to the $CO_3^{2-}$ stretching vibration, illustrates the distribution of

calcium carbonate (calcite) within the particles and is shown in yellow. The spectral map of the 1050 cm$^{-1}$ peak attributed to the liquid $NO_3^-$ stretching vibration provides the spatial distribution of water-solvated nitrate and is shown in white. The spectral map of the 993 cm$^{-1}$ peak assigned to sodium sulfate (thenardite) is reported in pink. The spectral map of the 1017 cm$^{-1}$ peak characteristic of calcium sulfate anhydrite (recrystallized sea-salt droplets) is depicted in cyan. Finally, the spectral map obtained at 1068 cm$^{-1}$, indicative of solid sodium nitrate (nitratine) is shown in green. The observed particles

consist mainly of $NaNO_3$ solid cores agglomerated with some amount of $Na_2SO_4$ (thenardite) surrounded by a liquid droplet containing $NO_3^-$ ions. These particles, marked with a white arrow in Fig. 12, were classified as sea-salts when observed by SEM/EDX. Numerous particles are also formed as a mixture of solid $NaNO_3$, $CaSO_4.2H_2O$ and liquid nitrate ion (an example marked by a red arrow in Fig. 12). They were classified as mixed dust/seasalt particles when analyzed by SEM/EDX. Generally, sodium nitrate particles partially recrystallize during analysis due to local heating under the laser

beam. The remaining particles probably contain Raman inactive NaCl and undetected species. Furthermore, Raman analysis is conducted at ~60% RH. Curiously, some particles remain with a droplet shape (marked by a green arrow in Fig. 12). This points to a probable presence of calcium nitrate with very low deliquescence RH in the range of 10-18% (Laskin et al., 2005a; Tang et al., 2016). Thus, these Ca-rich particles may have been collected as droplets. They were classified as dust particles when examined by SEM/EDX.




## 6 The impact of the sea breeze on the aerosol radiative effect

In this section, we evaluate the impact of the sea breeze on the broadband solar radiation, which occurs due to the perturbation of the aerosol properties. The diurnal variability of the solar radiative flux at the Earth's surface generally follows a monotonic and smooth curve as a function of time or solar zenith angle if the sky is clear and atmospheric

conditions are stable. Perturbations of the solar flux can appear due to presence of clouds or changes in aerosol characteristics. In Fig. 13a, we present the solar flux at the surface as a function of time, which is measured by the pyranometer of SolRad-Net for the afternoon of August 16. An irregular drop in the solar flux occurs at 14:00 UTC, which is the time of arrival of the sea breeze front. The discontinuity in the slope implies a loss of solar energy received at the surface presumably due to the changes in the aerosol properties. To evaluate this sea breeze induced radiative effect, we calculate the

solar fluxes and the net instantaneous direct aerosol radiative effect using a computational tool described in (Derimian et al., 2016). The calculations of the solar flux employ the aerosol models retrieved by AERONET and parameters of the gaseous concentrations and surface reflectance at the site for August 16 that are adopted from the database of the AERONET operational code. The results of the simulated solar flux are superimposed on the results of the measurements in Fig. 13b and are presented as a function of the corresponding solar zenith angles. The fluxes that are calculated for the aerosol

characteristics retrieved just before (red line) and during (blue line) the sea breeze are in good agreement with the measurements and the magnitude of the drop in the measured flux. Thus, the difference between the red and the blue lines, for the same solar zenith angle, corresponds to the loss of solar energy reaching the surface due to the sea breeze related anomaly in aerosol optical properties. For example, at a solar zenith angle of 60°, which corresponds to the time of the sea breeze front, the reduction of the solar flux is about 23 Wm$^{-2}$. This amounts to an about 5 % reduction of the total solar flux

that would reach the surface without the sea breeze effect. We now evaluate the aerosol instantaneous net direct radiative effect, which is defined as the difference between downwelling and upwelling fluxes at a given atmospheric layer in aerosol-free and aerosol-laden conditions. The instantaneous radiative effect refers to a value at a particular solar zenith angle. The radiative effect is formulated such that a negative sign signifies a radiative cooling. Thus, a negative value at the bottom of atmosphere signifies a radiative cooling at the surface. At the top of atmosphere, a negative value signifies additionally

reflected radiation due to aerosol presence and therefore a radiative cooling of the whole surface-atmosphere system. More details about the calculation tool and approach used can be found in (Derimian et al., 2016). Figures 12c,d present the calculated instantaneous net aerosol radiative effect at the bottom and the top of atmosphere before and during sea breeze. For example, before the sea breeze the background aerosol produces a radiative effect of up to about -10 Wm$^{-2}$ at the ground and -5 Wm$^{-2}$ at the top of atmosphere. Then, the negative aerosol radiative effect increases during the sea breeze up to -20.5

W m$^{-2}$ at the ground and -6.6 W m$^{-2}$ at the top of atmosphere. We can therefore estimate a doubling of the aerosol radiative cooling effect at the surface and an increase by almost one-third at the top of atmosphere due to the sea breeze effect on this specific day. The difference between the net top and net bottom radiative effects is the atmospheric radiative effect. It represents the part of the energy that is trapped in the atmosphere due to the aerosol presence. The atmospheric radiative



effect is always positive if the aerosol absorption is not zero and represents the radiative warming of the atmospheric layer. The atmospheric radiative effect increases during the sea breeze by almost three times, that is from about 5 W m$^{-2}$ before the sea breeze to about 14 W m$^{-2}$ during the sea breeze. We therefore can conclude that the sea-breeze-induced changes in the aerosol characteristics can lead to an important relative change in the background aerosol radiative effect.

**7 Aerosol core-shell structure and implication for remote sensing**

As follows from the individual particle analysis presented in Sect. 5, coatings of particles by a liquid layer are quite probable even in locations believed to be dominated by hydrophobic aerosols. At the same time, only a homogeneous particle model is used in remote sensing algorithms. Generally, the reason for this is a lack of sensitivity of the remote sensing measurements to detailed aerosol microphysical characteristics. In this section, we attempt to verify the possible impact of

the core-shell structure on the aerosol microphysical parameters retrieved using the homogeneous particle assumption in the AERONET operational retrievals. We also discuss the implications for another type of remote sensing measurements, motivated by the possibility that particles with complex microphysics can provide optical characteristics that are hard to reproduce using a homogeneous particle model. However, it is possible that the aerosol microphysical characteristics, retrieved using the homogeneity assumption, will be an equivalent that satisfies the radiative properties of more complex

microphysics. The question that we therefore examine is: how can the core-shell structure affect the retrieved aerosol spectral complex refractive index, volume size distribution, and fractions of spherical-nonspherical aerosols, if a homogeneous particle model is assumed in the retrievals? It should be mentioned here that, with respect to the AERONET retrievals, Dubovik et al., (2000) already provided a test of the effect of internal (core-shell) mixture on the retrieved aerosol microphysical parameters using a simple model of black carbon core and water-soluble substance shell. Tests were

performed also for the possible effects of external mixing and the assumption of aerosol sphericity as part of the accuracy assessment of aerosol optical properties retrievals from AERONET. It is noteworthy that, because the aerosol sphericity assumption was found to cause artifacts, the randomly oriented spheroids model was introduced in the retrieval algorithms (Dubovik et al., 2006). However, the tests of Dubovik et al., (2000) for the influence of external and internal (core-shell structure) aerosol mixture on the retrievals did not show anomalies in the retrieved size distribution, while the retrieved real

and imaginary parts of the complex refractive index yielded equivalent values that were generally in between the refractive indexes of the components constituting the mixture.

In the current study, we calculate the directional aerosol optical properties of homogeneous and core-shell particles that reflect our observations in the Negev Desert. They are then inverted using the same inversion scheme as AERONET in order to verify the applicability of the conclusions in (Dubovik et al., 2000) to our case study. We also analyze a case where the

phase function across the full angular range is available for the retrievals. Note, that the calculations presented here are performed in a single scattering approximation and not for radiances as they would be observed by a sun/sky photometer and




as presented in (Dubovik et al., 2000). The reasoning is that if the differences are not significant in a single scattering case, then they will be diminished even more in the case of multiple scattering under real atmospheric conditions.

Figure 14a,b presents the spectral aerosol optical thickness and the directional distribution of scattered light intensity ($P11(\theta) \cdot AOT_{scat}$) that are further used for the inversion employing a conventional homogeneous particle model;

panel c illustrates the directional distribution of the degree of linear polarization ($-P12(\theta)/P11(\theta)$) of the scattered light. $AOT_{scat}$ is the scattering aerosol optical thickness, $P11(\theta)$ and $P12(\theta)$ are the elements of the scattering matrix, where $P11(\theta)$ fulfils the normalization condition of

$$\frac{1}{2}\int_0^\pi P11(\theta) \cdot sin\theta d\theta = 1 \qquad (2)$$

Three simplified scenarios are considered: first - particles are homogeneous, second and third – a liquid water layer coats particles with a thickness that corresponds to 10 % and 40 % of the total particle radius, respectively. It is important to note that the total particle radius is kept constant in all three scenarios in order to rule out effects of changing aerosol size distribution. In the case of the coated particles, the size of the core is therefore proportionally decreased. The spectral AOT

presented in Fig. 14a is normalized to the maximum value in order to show the variability in spectral dependence due to the coating. A change in the spectral AOT and angular dependence of $P11(\theta) \cdot AOT_{scat}$ (Fig. 14 a and b) is present for the case of 10 % and is visibly significant for the case of 40 % shell thickness. In order to evaluate the impact on the retrieved microphysical parameters, we now invert simultaneously the spectral AOT and $P11(\theta) \cdot AOT_{scat}$. Note that $P11(\theta)$ is calculated for scattering angles from 0 to 180 degree and with a resolution of 1 degree, which represents an ideal possible

scenario of measurements. The wavelengths employed for the spectral AOT and $P11(\theta) \cdot AOT_{scat}$ are 440, 670, 870 and 1020 nm, which are the operational wavelengths of the AERONET retrievals. The assumed complex refractive index of the core is 1.47+0.003$i$, based on the values obtained for the aerosol model before the sea breeze installation; it also assumed to be spectrally independent for the simplicity.  The assumed complex refractive index of the shell is 1.33+0.0$i$, which is the value for pure water and is also assumed to be spectrally independent.

For the case when the forward calculations are conducted using the homogeneous aerosol model, the inversion procedure reproduces very well the assumed aerosol size distribution, the real and imaginary part of the complex refractive index, and the fraction of spherical particles (Fig. 15, homogeneous case). Note that the refractive index used in the case of homogeneous particles is the same as that of the core. The homogeneous scenario is an initial and necessary test, which illustrates first of all the consistency of the calculations, and second, that accurate characteristics can be retrieved having

spectral AOT and full angular range phase function. For the case when the forward calculations are conducted for 10 % coating thickness, the retrieved real refractive index is greater than that of the core, a discrepancy appears in the sphericity fraction, and the residual of the fit increases (see insert table in Fig. 15). The most important disagreement appears in the case of 40 % coating thickness. The retrieved refractive indexes significantly exceed those of the core, and a quite different





size distribution is required in order to fit the spectral AOT and phase function using a homogeneous particles model. Note that the residual error of the fit is as high as of 14 %, in contrast to 1 % or 1.9 % in the two previous scenarios, which indicates the difficulty of an accurate reproduction of the spectral AOT and full angular range phase function. It follows that, at least for the case considered here, the core-shell particle structure can have characteristics that are difficult to reproduce by

an equivalent homogeneous aerosol model. As can be seen from Fig. 14b), the main differences in the phase function of a core-shell aerosol relative to a homogeneous one are in the backward scattering. However, ground-based photometers cannot observe backward scattered light and measure the radiation scattered mostly in the forward direction; up to about 120 degree of the scattering angle, depending on the sun elevation angle during almucantar measurements. In order to mimic the AERONET angular observations range, we conduct calculations for the phase function in the angular range from 0 to 120

degrees. Evidently, by limiting the angular range we lose sensitivity and, in the case of homogeneous particles, the real and imaginary refractive indexes are now not retrieved as well as in the case of the full range of the scattering angles (red line in Fig. 16b, c versus in Fig. 15b, c). However, the values are still comparable to the originally assumed values, i.e., those of the core. The scenario of 10 % coating thickness provides results very similar to the homogeneous case. A difference, however, appears for the scenario of 40 %, when the real refractive index is clearly in between the refractive indexes of the dust core

and water shell components. It is noteworthy that the result is consistent with the tendency observed at the Sede Boker site that the real refractive index decreases as the water vapor concentration increases. Also, notable is appearance of a similar spectral dependence of the real refractive index as in the case of the sea breeze, i.e., lower values at shorter wavelengths. This can be due to a stronger sensitivity of the radiation at shorter wavelengths to the shell material on the surface of the particle, whereas radiation at longer wavelengths is more influenced by the internal part of particle. The imaginary refractive

index becomes greater than the one of core, a tendency that is also visible in the observations. The residual of the fit is quite high, which means that a physical interpretation of the retrieved microphysical parameters should be done with caution. However, the positive side of the obtained high fit error is that it shows the possible sensitivity of the measurements to the core-shell structure. That is, it appears that a homogeneous particles model is not able to reproduce accurately the characteristics of the core-shell structure. Similar conclusions can also be made for the case of the inversion of the phase

function over the full angular range. Additionally, Fig. 14c presents a not yet discussed variability in the degree of linear polarization. This deviation of the degree of linear polarization from the homogeneous particle scenario is even stronger than that of the phase function. For instance, there is even a sign reversal in the peak at a scattering angle of about 170 degree in the case of 40 % coating thickness. The results of our tests (not shown here as graphs) of a simultaneous inversion of spectral AOT, $P11(\theta) \cdot AOT_{scat}$, and $-P12(\theta)/P11(\theta)$, show similar tendencies of increasing the residual error of the fit and

aberrant refractive index and size distribution as the thickness of the coating increases. Thus, even stronger sensitivity to the core-shell structure is expected if polarization is measured.

It can be concluded that, in some measurement configurations, an equivalent homogeneous particle model can indeed represent the optical characteristics of a liquid coating even under the assumption of a single scattering approximation, which is in general agreement with the results of (Dubovik et al., 2000). However, we can also conclude that including backward





scattering angles and polarimetric measurements can bring about more sensitivity to the core-shell structure, and thus more systematic studies are needed. Particular attention should be paid to satellite observations, because the main differences due to the aerosol core-shell structure are observed in the angular and polarimetric characteristics of the backward scattered light; important implications for LIDAR measurements are also possible.

**8 Conclusions**

The influence of the sea breeze on the atmospheric aerosol chemical composition, microphysical, optical and radiative characteristics in the Negev Desert of Israel during summer time is shown for the first time and discussed in details. We employed extensive remote sensing observations, *in situ* aerosol sampling and laboratory physico-chemical characterization of the particles. We found that at an arid location at a distance of at least 80 km from the Mediterranean seashore, marine

aerosol particles and air masses are influencing daily the desert aerosol composition. While the entire phenomenon lasts for about three hours, an abrupt increase and a peak in aerosol size, volume concentration, and optical thickness, as derived by AERONET observations in the solar spectrum, occur at the front of the sea breeze arrival. At the same time, the sky brightness temperature, derived by radiometric measurements in the thermal infrared spectrum, also increases and shows a weaker spectral dependence, which indicates a contribution of large particles or water droplets (the former behaves like a

black body in TIR). The effect of the sea breeze front on the atmospheric radiative characteristics was most obvious in the measurements by the thermal infrared radiometer. This illustrates the great potential of the simultaneous and complementary observations of solar and thermal infrared radiances for aerosol studies.

We found that the fraction of hygroscopic marine and internally mixed dust/marine aerosol particles increases significantly during the sea breeze; however, similar particles are present as a background in the Negev Desert also before the sea breeze

arrival. The increase in atmospheric water vapor and relative humidity is associated with the sea breeze arrival and the aerosol size distributions show a shift towards larger particles. We suggest that hygroscopic growth can explain the observed shift in the aerosol size distribution. This hypothesis is supported by SEM/EDX analyses, which show that a large number of particles are surrounded by liquid residuals. Despite the desert location of the site, we also found that a large fraction of the sampled particles is composed of highly hygroscopic material. Although particles of all sizes grew, the size shift of the fine

mode was stronger (see Fig. 6). This can be linked to the size-dependent aerosol composition, which shows higher fractions of hygroscopic particles in PM1, consisting of marine and internal dust/marine mixtures. We also observed that a large number of dust particles had a liquid coating in the form of water-solvated nitrate. These nitrates have an anthropogenic origin and their internal mixture with dust, namely Ca-rich particles, makes them highly hygroscopic; the deliquescence RH of such particles can be as low as about 10 or 20%. Thus, even in a dry desert environment, such dust particles can have a

substantial liquid content, as was observed also in our samples. The sampling results are also in line with the decreasing values of the retrieved by AERONET real refractive index, which indicates the presence of water in the aerosol composition. The observed liquid coating of particles can have more general implications for the modeling of the aerosol scattering and





absorption properties. This is because all present-day remote sensing algorithms for retrieval of aerosol microphysical properties assume homogeneous particles. Indeed, there are practical reasons for this assumption, which are related to issues of measurement sensitivity, as discussed in Sect. 7. Based on the numerical simulations presented here we suggest that a sensitivity of remote sensing to the core-shell structure exists in those observational configurations where information about

the phase function in an extended angular range and polarimetric measurements are used, and that scattering in the backward directions is particularly important.

In addition to the individual particle microphysics, the sea breeze also perturbs the radiative budget. On a specific day, August 16, 2012, we estimated a 5 % reduction in the broadband solar radiation reaching the surface due to the sea-breeze-induced change in the aerosol characteristics. The background net aerosol radiative cooling was doubled at the surface and

increased by about one-third at the top of atmosphere. The atmospheric radiative warming, which is the difference between the net top and net bottom radiative effects, increased by about a factor of three. The cooling of the surface and warming of the atmospheric layer can change the gradient of the atmospheric temperature profile, which may imply a feedback on the sea breeze dynamic; this subject merits a separate dedicated study.

This study illustrates the complexity of the aerosol microphysics when marine, desert, and pollution air masses interact. An

understanding and proper modeling of aerosol optical properties in coastal areas should be of high importance because densely populated and industrial centers tend be located on the seashores. The sea breeze occurs in many locations around the world and this systematic phenomenon can be used as a natural laboratory to study and evaluate the impact of the aerosol mixing state and hygroscopicity on aerosol optical properties and radiation.

**Acknowledgments**

The CaPPA project (Chemical and Physical Properties of the Atmosphere) is funded by the French National Research Agency (ANR) through the PIA (Programme d'Investissement d'Avenir) under contract "ANR-11-LABX-0005-01" and by the Regional Council " Nord Pas de Calais - Picardie» and the "European Funds for Regional Economic Development (FEDER). The SEM facility in Lille (France) is supported by the Conseil Regional du Nord-Pas de Calais, and the European Regional Development Fund (ERDF). The MPLNET project is funded by the NASA Radiation Sciences Program and Earth

Observing System. AL acknowledges support from the W.R. Wiley Environmental Molecular Sciences Laboratory (EMSL), a national scientific user facility located at PNNL, and sponsored by the Office of Biological and Environmental Research of the U.S. DOE. PNNL is operated for US DOE by Battelle Memorial Institute under Contract No. DEAC06-76RL0 1830. We thank the MPLNET PIs Ellsworth Judd Welton and Sebastian A. Stewart for their effort in establishing and maintaining the Sede Boker site. The authors gratefully acknowledge the NOAA Air Resources Laboratory (ARL) for the provision of the

HYSPLIT transport and dispersion model and the READY website (http://ready.arl.noaa.gov) used in this publication. Special thanks to Mr. Alexander Goldberg from The Jacob Blaustein Institute for Desert Research, Ben Gurion University, Sede Boker Campus for highly valuable technical help, which enabled the proper functioning of the instrumentation at the



site. We also thank Prof. Abraham Zangvil and Mr. David Klepach from the same institution for providing the
meteorological data.

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



| Date: August 16, 2012 | Sample 1 (S1): Before sea breeze | Sample 2 (S2): During sea breeze |
|---|---|---|
| **Start time** | 13:00 UTC | 14:30 UTC |
| **Duration** | 60 min | 15 min |
| **RH (%)** | 28.5 | 52.0 |
| **Air temperature (°C)** | 32 | 30 |
| **Wind speed (ms$^{-1}$)** | 4.0 | 7.3 |
| **Wind direction (°)** | 306 (NW) | 308 (NW) |

**Table 1. Sampling conditions. Relative humidity, air temperature, wind speed and direction are the mean values during the sampling time.**

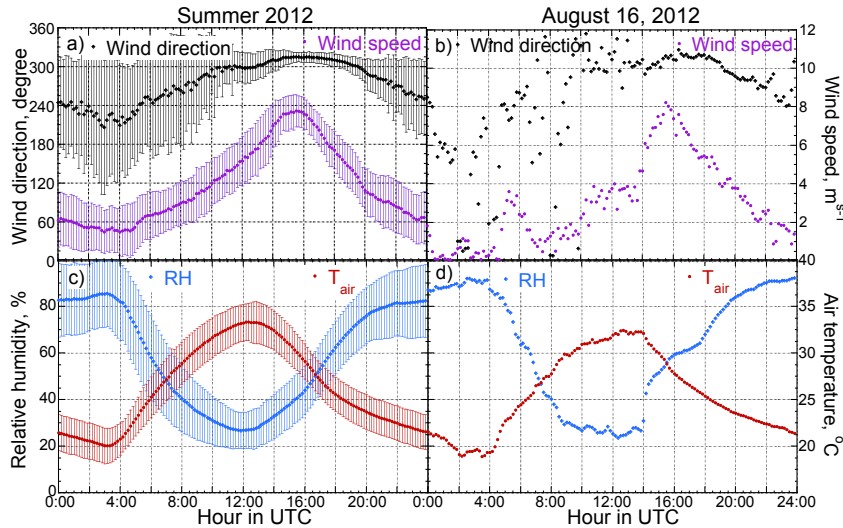

**Figure 1. Mean diurnal variability of (a) wind direction and speed, (c) relative humidity and air temperature calculated from three months (June, July, August 2012); error bars correspond to ± one standard deviation. Panels (b) and (d) show the diurnal cycle of the same variables, but for August 16, 2012.**



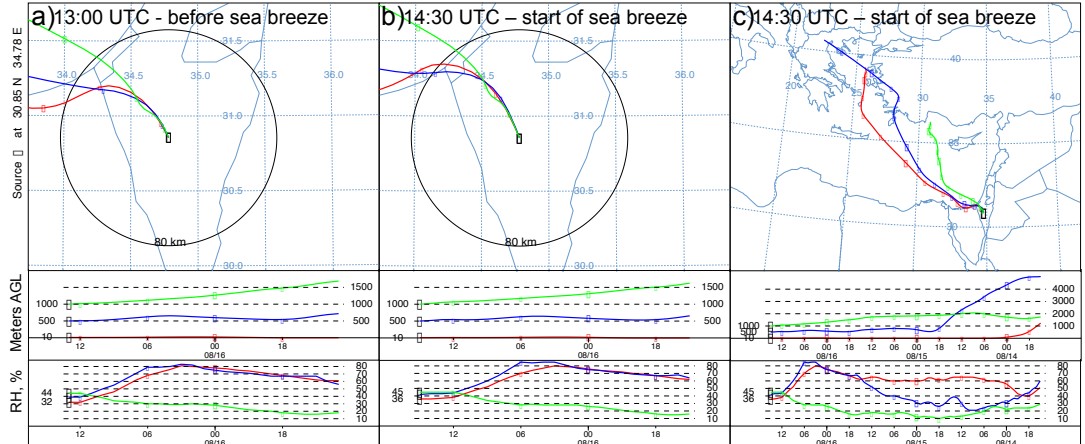

**Figure 2. (a) and (b) 24-hour and (c) 3-day backward trajectories ending at 13:00 and 14:30 UTC for altitudes above ground level (AGL) of 10 m (in red), 500 m (in blue) and 1000 m (in green) at the Sede Boker site; corresponding relative humidity along the trajectories are also presented.**

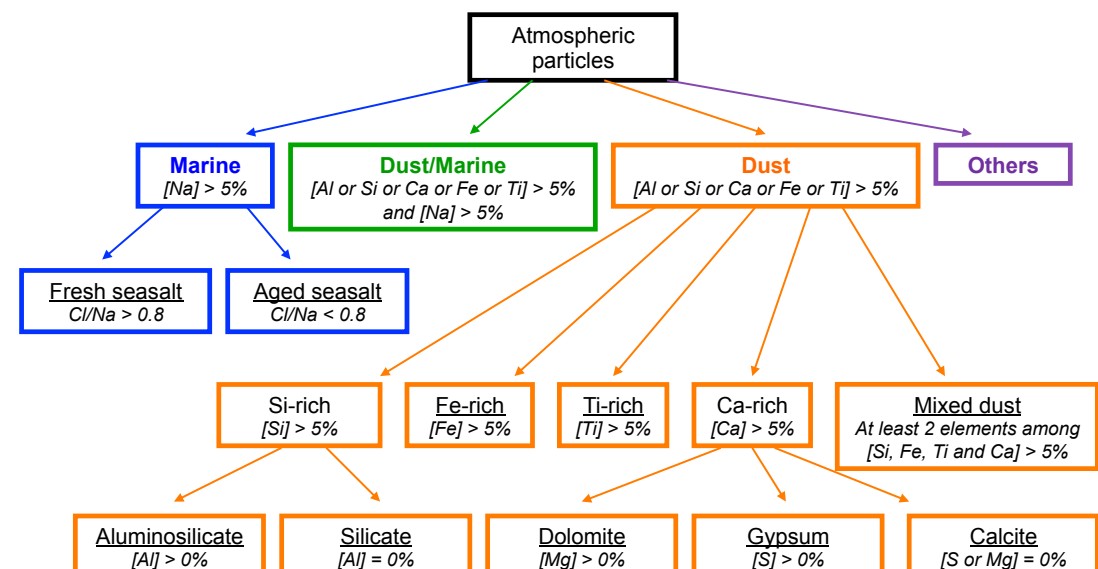

**Figure 3. Particle classification (colored text) and identification (underlined text) based on normalized atomic percentages for elements with Z > 10.**



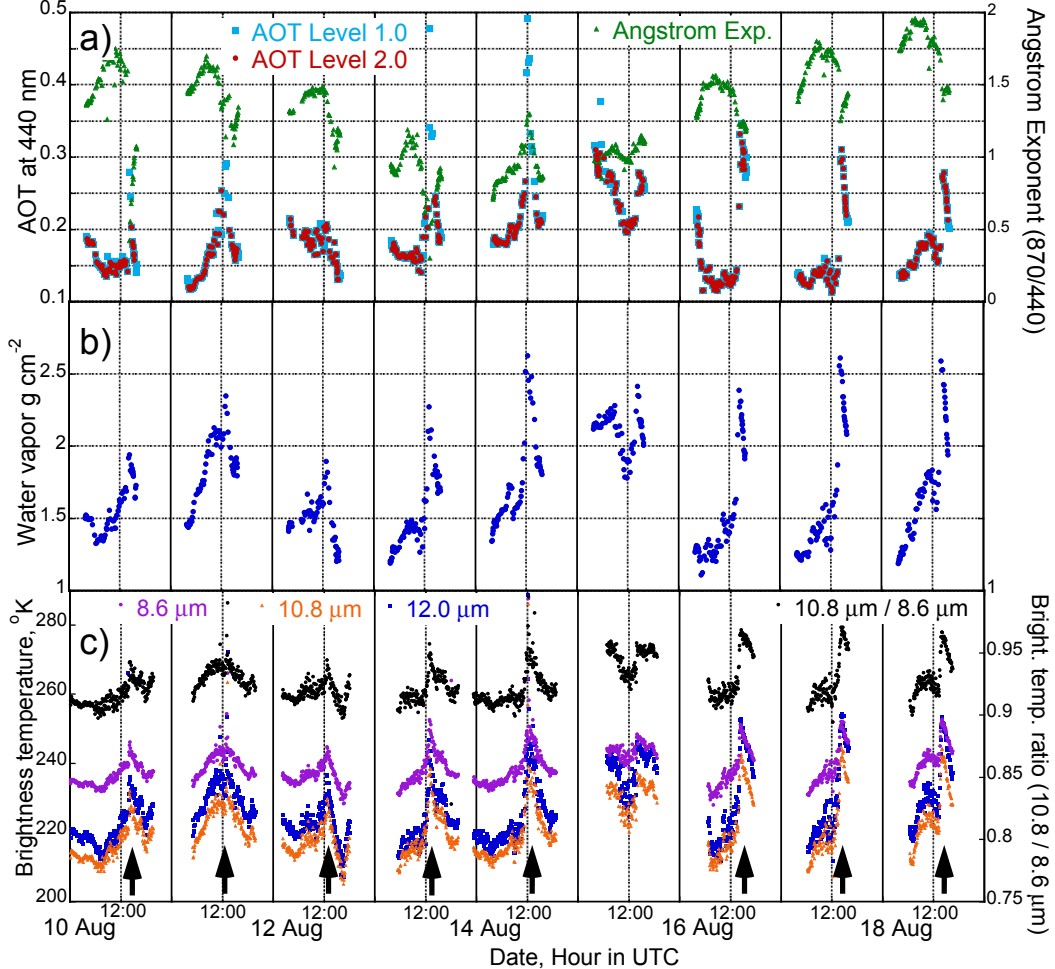

**Figure 4.** Time series of (a) AERONET observations of AOT at 440 nm and Ångström exponent between 870 nm and 440 nm, (b) AERONET derived total column water vapor and (c) sky brightness temperature as measured by the thermal infrared radiometer at three spectral channels. Arrows indicate the signal peaks corresponding to the sea breeze arrival occurred on eight of the nine days presented.





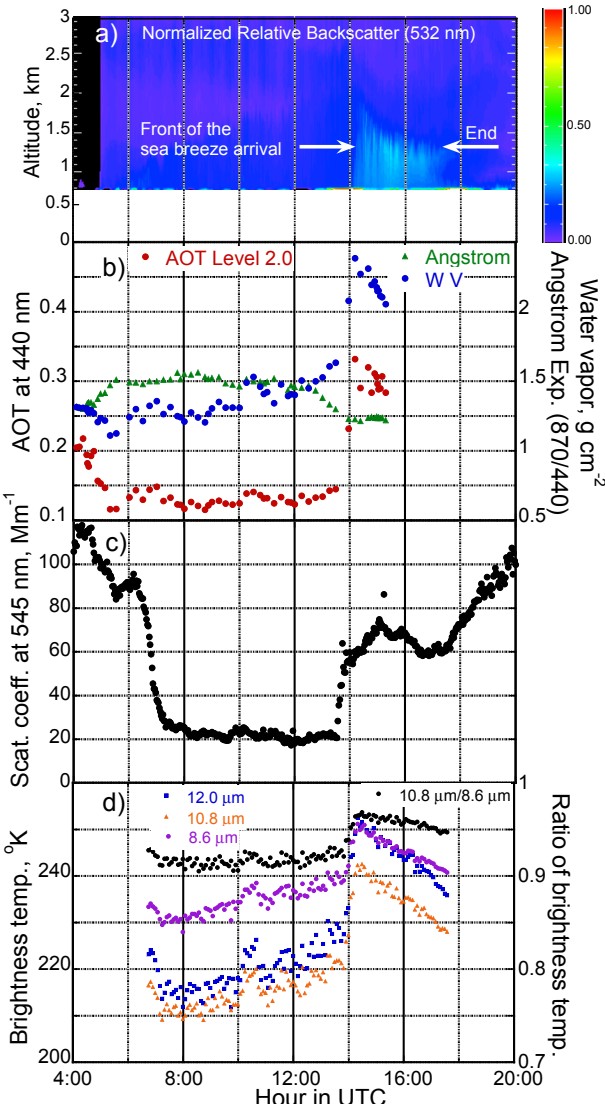

**Figure 5.** Diurnal variability for August 16, 2012 of: (a) vertical distribution of LIDAR backscatter signal at 532 nm (it starts from about 700 m since observations are generally omitted in the first lower hundreds meters); (b) AOT at 440 nm (level 2.0 – after the cloud screening and quality assurance), Ångström exponent between 870 and 440 nm, and water vapor; (c) scattering coefficient at 545 nm and ambient RH; (d) sky brightness temperature of the thermal infrared radiometer in channels centered at 8.6, 10.8, 12.0 mm and the brightness temperature ratio between 10.8 mm and 8.6 mm.




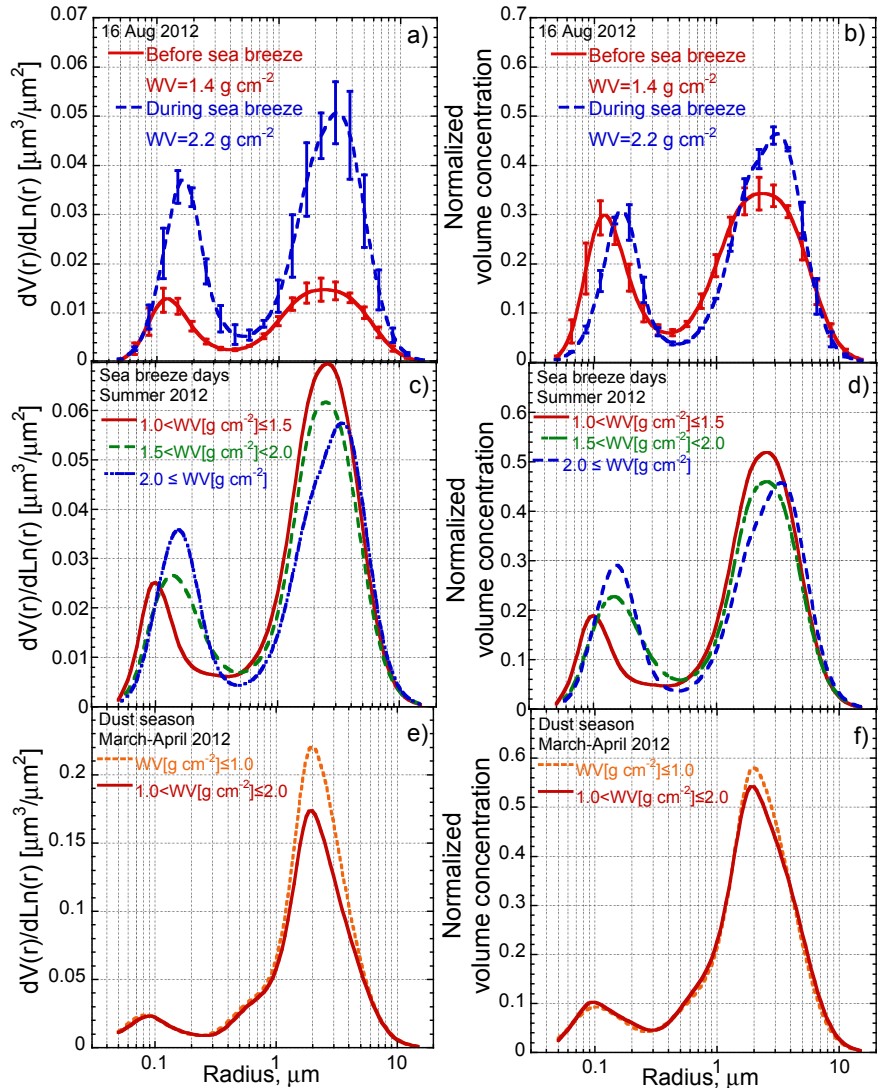

**Figure 6.** The left column shows aerosol volume size distributions and the right column shows the normalized to total volume size distributions. (a) and (b) for August 16, 2012; (c) and (d) present averaged size distributions for the sea breeze days of three summer months, 2012; and (e) and (f) are for the dust period (March - April) of 2012. Error bars in (a), (b) correspond to the standard deviation; the error bars in (c)-(f) overlap and are not shown for clarity of the figure.



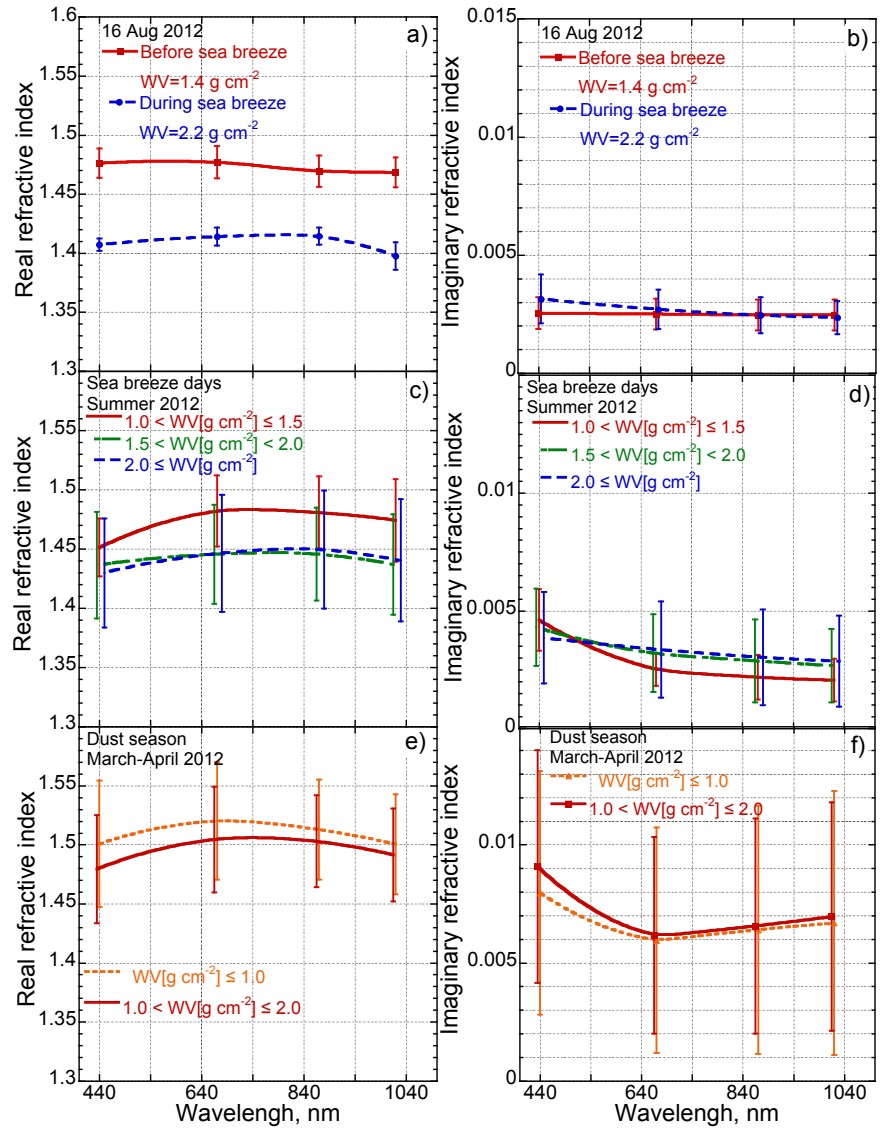

**Figure 7.** The left column shows the real part and the right column shows the imaginary part of averaged values of complex refractive index as retrieved by AERONET. (a) and (b) for August 16, 2012; (c) and (d) for sea breeze days of three summer months, 2012; and (e) and (f) for the dust period (March - April) of 2012. Error bars correspond to the standard deviation.




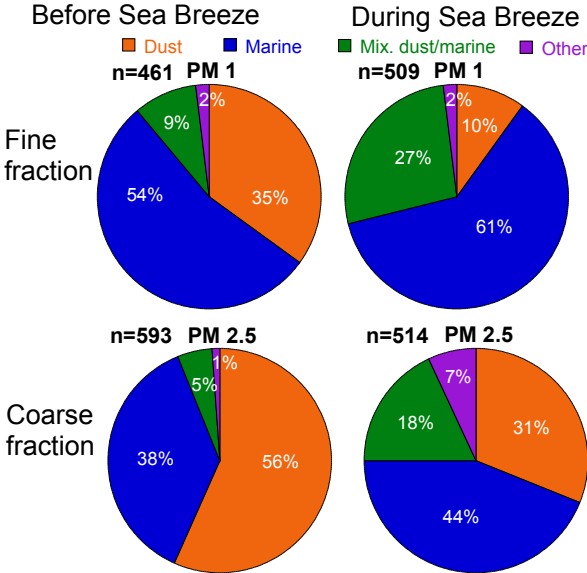

**Figure 8.** Relative proportions of particle types obtained from CCSEM/EDX analysis of particles collected (left) before the sea breeze and (right) during the sea breeze. Fine and coarse fractions correspond to the aerodynamic cut-off diameters of the cascade impactor of 1 μm and 2.5 μm, respectively. Particles are sorted into the following four main groups: Marine, Dust, Mixed Dust/Marine and Other. The total number (n) of the analyzed particles is also indicated in each panel of the figure.

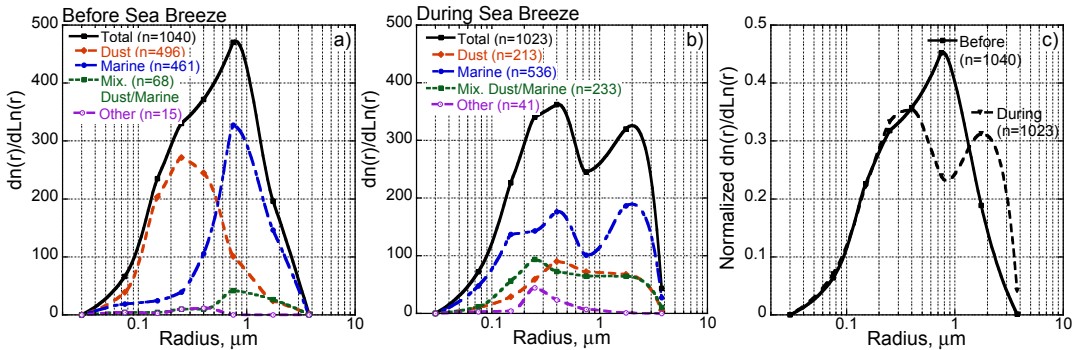

**Figure 9.** Number size distributions of particles analyzed by CCSEM/EDX – (a) before the sea breeze, (b) during the sea breeze. (c) Size distributions normalized to the total number of particles analyzed for before and during the sea breeze.





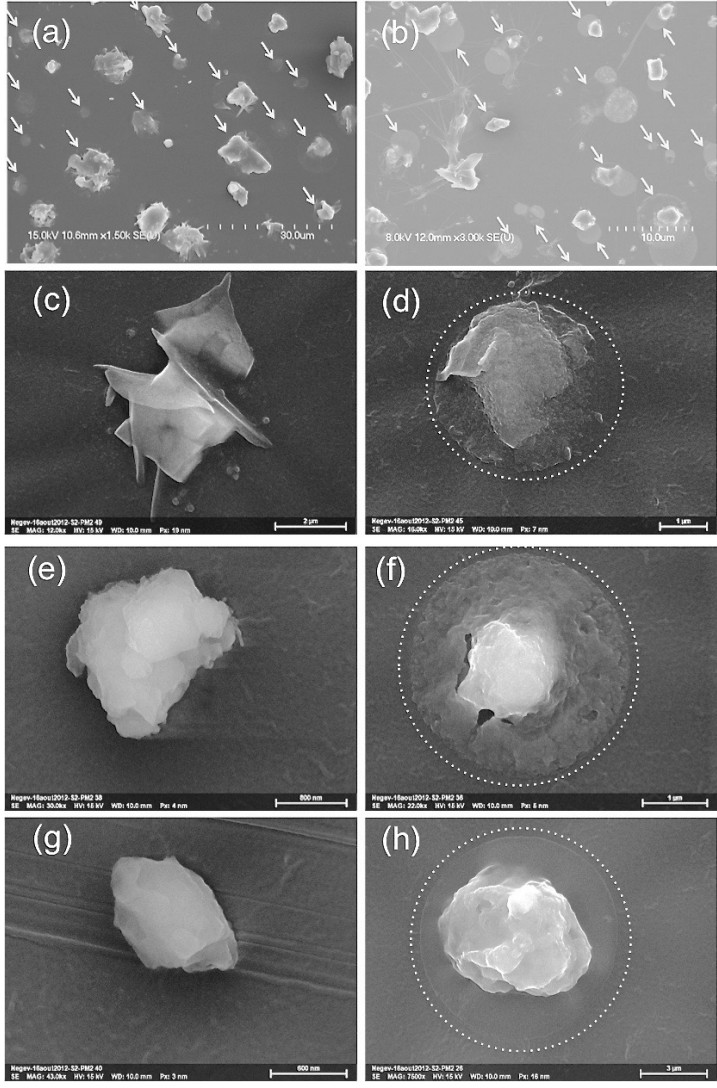

**Figure 10. Secondary electron images of (a) particles of the coarse fraction, (b) particles of the fine fraction, (c-h) individual particles typical of: c) fresh marine, (d) aged marine, (e) unreacted dust (silicate), (f) internally-mixed dust/marine, (g) unreacted dust (calcite), (h) aged Ca-rich dust (calcite partly converted to calcium nitrate). Arrows mark the presence of halos. Dashed circles depict the boundaries of halos.**



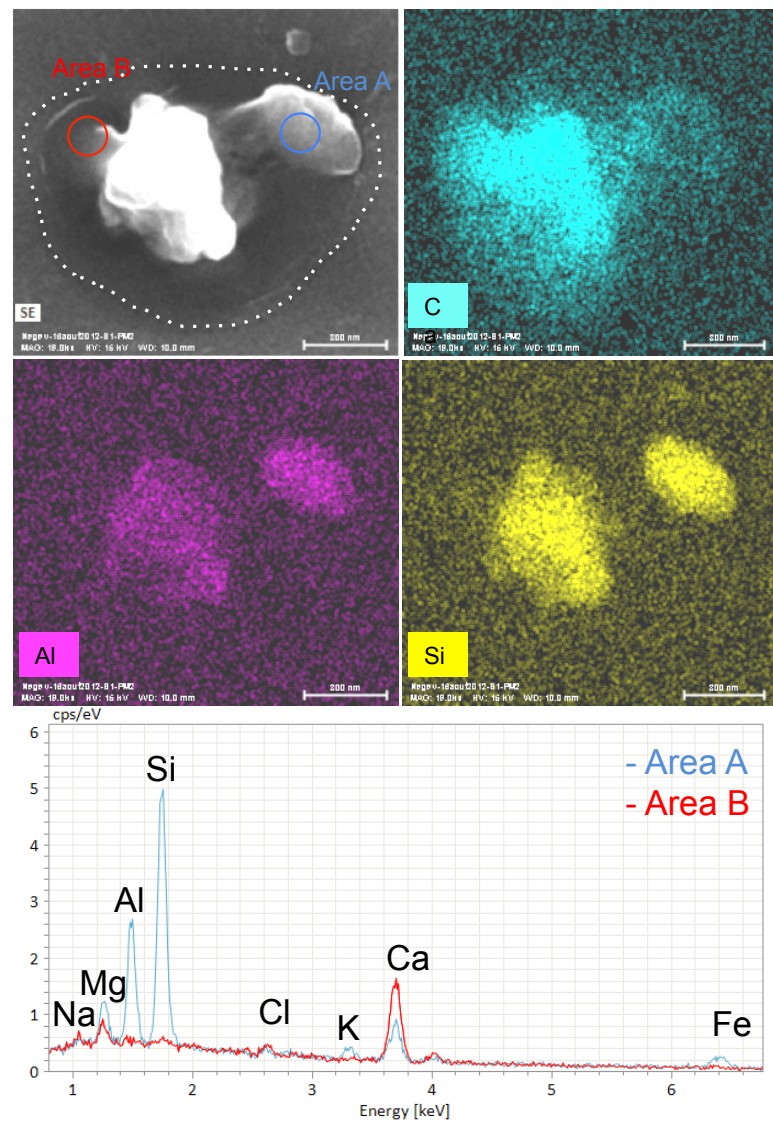

**Figure 11. Secondary electron image, elemental energy dispersive X-ray (EDX) mappings (Al, Si and Ca) and EDX spectra of an individual internally mixed calcium nitrate/aluminosilicate particle. Scale marker bars correspond to 800 nm. Dashed line depicts the boundaries of a halo.**





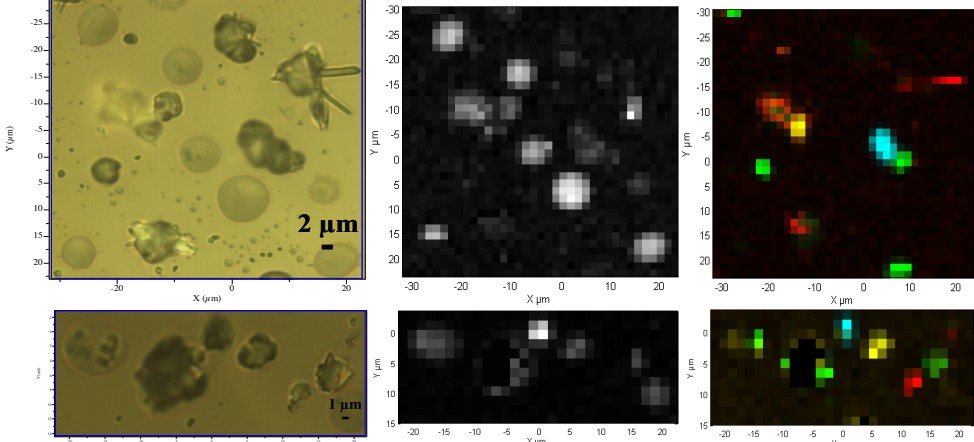

**Figure 12. Optical images (left panels) of coarse particles collected before (top panels) and during (bottom panels) sea breeze and corresponding Raman molecular mappings (middle and right panels). Raman maps are colored according to the band intensity at**
5 **$1050$ cm$^{-1}$ (middle panel), and $1068$ cm$^{-1}$, $1086$ cm$^{-1}$, $1017$ cm$^{-1}$, $993$ cm$^{-1}$ (right panel), respectively. White: water-solvated nitrate ion; Green: solid sodium nitrate (nitratine); Yellow: calcite; Cyan: calcium sulfate anhydrite; Pink: solid sodium sulfate (thenardite). The meaning of the colored arrows is described in the text.**





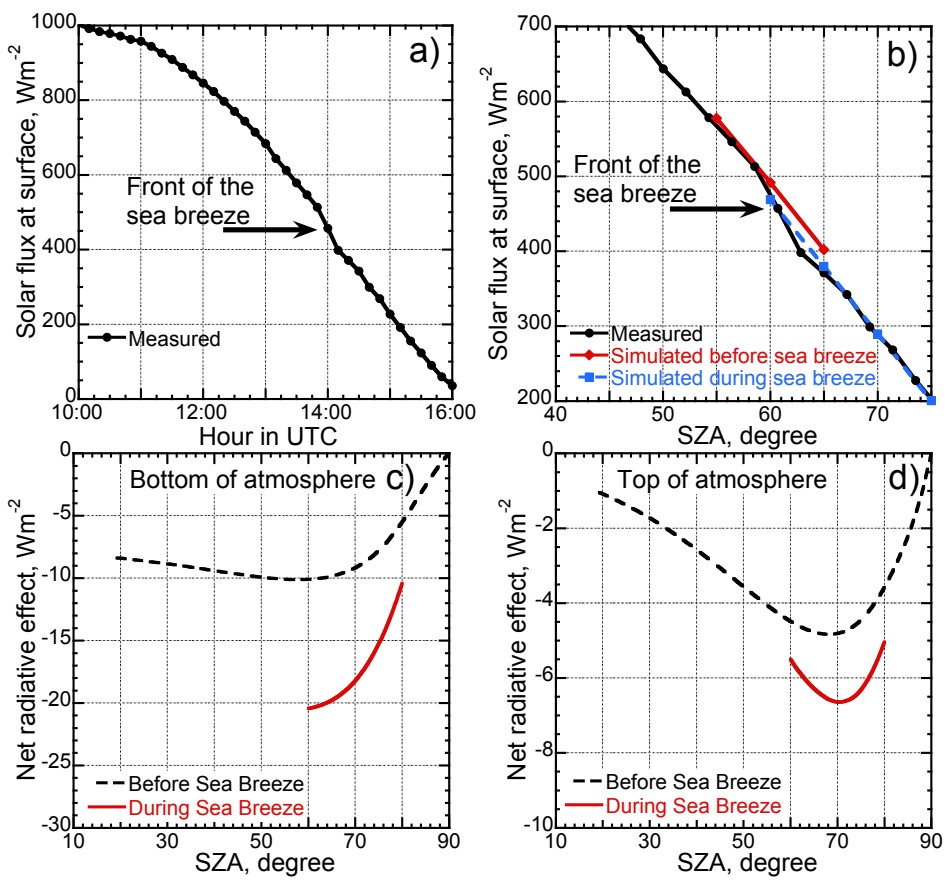

**Figure 13. (a) Solar flux measured at the surface by the pyranometer of SolRad-Net in the afternoon of August 16. (b) Measured solar flux and results of the flux simulations based on the AERONET retrievals of aerosol characteristics just before and during the sea breeze. (c) and (d) Net aerosol radiative effect calculated before and during the sea breeze at the bottom and the top of atmosphere. The measured flux in panel (a) is presented as a function of time and in panels (b) to (d) as a function of the corresponding solar zenith angles (SZA).**




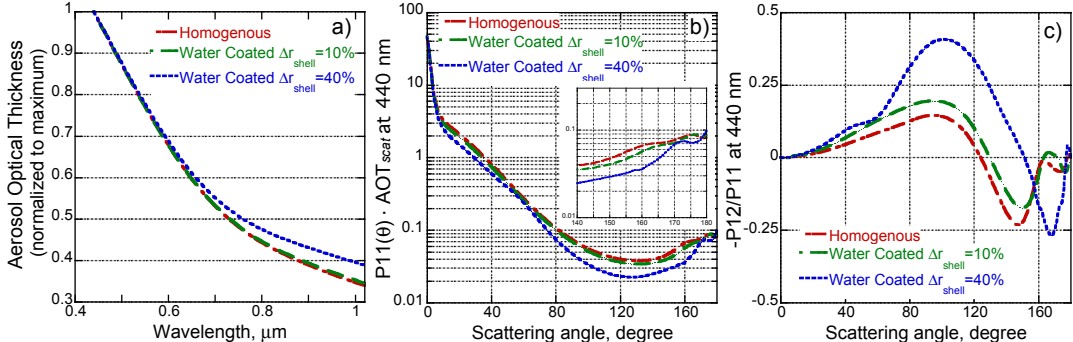

**Figure 14. (a) Spectral aerosol optical thickness (AOT), (b) phase function at 440 nm, and (c) degree of linear polarization at 440 nm calculated under the assumption of spherical homogeneous and coated particles, where the coating thickness is 10 and 40 % of the total particle radius. The characteristics are calculated for an aerosol size distribution observed during the sea breeze and**
5 **complex refractive index of core and shell as described in this section; the refractive index for the homogeneous case is equal to that of the core. The presented AOTs are normalized to the maximum value.**

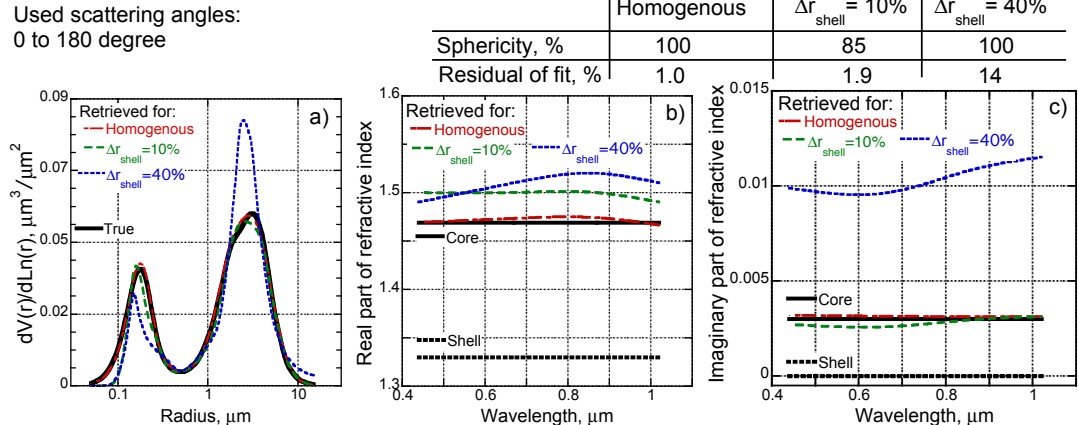

**Figure 15. (a) Aerosol size distributions, (b) real and (c) imaginary part of the complex refractive indexes as assumed in the**
10 **forward calculations of the aerosol optical characteristics (labeled "True", "Core", "Shell") and as a result of the inversion of the optical characteristics of homogeneous and core-shell particles presented in Fig. 14. The insert table summarizes the retrieved percentage of spherical particles and residual error between the assumed and the fitted aerosol optical characteristics. The results are obtained for the case when the scattering angle of the phase function ranges from 0 to 180 degrees.**





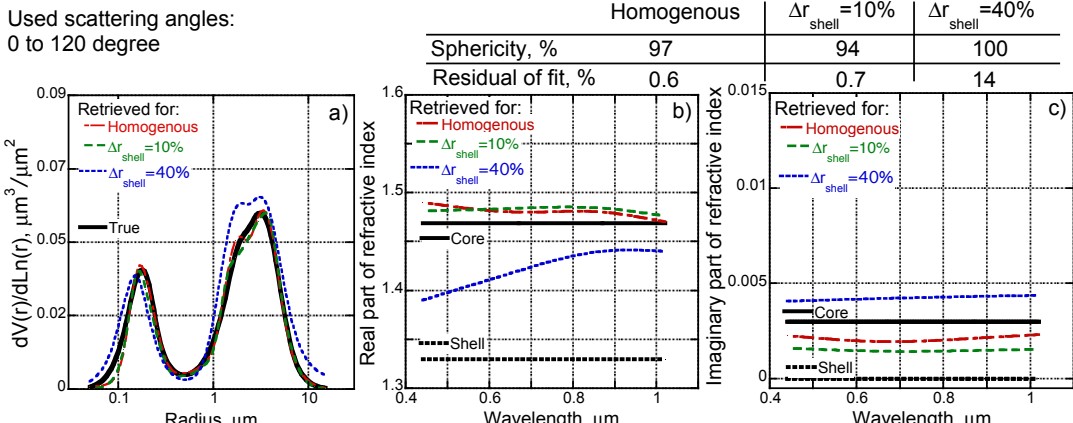

**Figure 16. Same as Fig. 15, but for a range of scattering angles from 0 to 120 degrees.**