# Peer review of "Effect of sea breeze circulation on aerosol mixing state and radiative properties in a desert setting"

_Atmospheric Chemistry and Physics, 2016_

## Referee Comment (RC1) · Anonymous Referee #3 · 17 Feb 2017

Review of 'Effect of sea breeze circulation on aerosol mixing state and radiative properties in a desert setting' by Derimian et al.

The paper studies the modifications in summertime sea breeze conditions of the aerosol compositional, microphysical, optical, and radiative properties at an inland location of the Negev Desert (Israel). It is well written and easy to read. The study is original and the scientific methodology sound. The complementarity of the inversion of remote sensing (sunphotometer) observations and of the direct, and I suppose time-consuming, off-line individual analysis for characterizing the particles is particularly interesting. The authors evidence for the first time the significant influence of the daytime intrusions of marine air on the aerosol characteristics at such a remote place.

[Figure]

Not only the composition, but also to internal structure of the particles greatly differ in the pre- and post- see breeze situations. In particular, the proportion of mineral (desert-dust) particles surrounded by a coating is unexpectedly large in both cases, which contradicts the common assumption that desert dust is hydrophobic. These modifications of the aerosols characteristics need to be taken into account for quantifying their radiative effect. Finally, the numerical simulations made by the authors show that the current remote sensing inversion algorithms need to be modified in order to take the core-shell structure of the particles into account. My opinion is that this paper deserves publication provided the following concerns are addressed: General comment: After reading the paper, one is left with the impression (see for instance lines 21-23, page 2) that the aerosols initially present at Sede Boker are modified by the arrival of the sea breeze. In fact, these pre-existing aerosols are most probably blown further downwind of the experimental site by the breeze and replaced by new freshly advected particles. The authors make an exhaustive and quite interesting comparison of the characteristics of these two sets of particles, but if the particles are not the same, is it possible to conclude that the size increase observed after the arrival of the sea breeze could be due to the water vapor uptake? More generally, the mineral particles observed during the marine intrusions probably have a long history of coexistence with the other species (sea-salt and anthropogenic aerosols and gases), what's more in humid air-masses. Therefore, they are more liable to have formed internal mixtures than the resident aerosol of Sede Boker. Miscellaneous: 1) P. 7: What can the origin of the Ti-rich particles be? 2) P. 8, line 25; on Fig. 4a, the Angström exponent increases with the arrival of the sea breeze on 14 August. Is there a plausible explanation for this exception to the rule? 3) P. 10, line 8: the nephelometer 'dries'... 4) P.10, lines 12-13: couldn't the 'abrupt response' also be due to the increase of the aerosols concentrations and to a shift in their size? 5) P.10, line 24: This is Fig. 6 (not 5) 6) P.10, line 33: The unit is $\mu$m not mm 7) Fig 6c and d : the blue line corresponds to WV larger than 8) P.11; line 5: As said in the following sentences, the large standard deviation does not allow concluding that there is 'a decrease' of the mean real refractive index. I would

remove the 'Curiously a decrease...'. 9) P.11; line 31 and the notations of Fig. 8: Usually, PM1 and PM2.5 correspond to particles with diameters smaller than 1 and 2.5$\mu$m, respectively. Here, PM1 corresponds to particles with diameters between 1 and 2.5, and PM2.5 to the range 2.5-10$\mu$m. This is confusing. 10) P. 12, lines 15-16: the 'other' particles represent 7% of the coarse fraction but are said to be smaller than 1$\mu$m in diameter. Isn't this contradictory? 11) P. 13, line 1: the authors say that the shift towards larger sizes of the marine particles during the sea breeze could be due to hygroscopic growth. Aren't the SEM observations made under a vacuum, i.e. in dry conditions? Moreover, if we go back to my first comment, please consider that the marine particles observed at the inland site before the arrival of the sea breeze might be more aged than the new ones. Consequently, their size-distribution might have been modified by the size-selective dry deposition process. For instance, I cannot help observing on Fig 9 that the very fine and the coarse particles present in the fresh marine air-masses (Fig. 9 b) have disappeared on Fig 9a, and that on the latter figure, only the particles with a diameter corresponding to the smallest deposition velocity (around 1$\mu$m) subsist. 12) P. 13, line 24: There is no Mg in the composition of the calcite. 13) Fig. 11: On the upper right-hand panel, this should be Ca (not C). 14) Fig. 12: I cannot see the arrows mentioned on page 14. 15) P. 15, line 19: In the reduction of 5%, what are the respective shares of the 1) aerosol changes and 2) WV increase? 16) P. 15, line 26: this should be Fig 13, not 12. 17) Section 7: please, consider reformulating the whole section. It is much harder to follow than the rest of the paper. For instance, the reader discovers only on page 17 that forward calculations have been made (and with which inputs), then that different scenarios have been considered for inversion simulations. 18) P.19, lines 2-5: could you be more specific regarding implications for the satellite and LIDAR inversions?

---

## Referee Comment (RC2) · Anonymous Referee #1 · 3 Mar 2017

A combination of remote sensing observations, in-situ measurements and chemical analysis at the individual particle scale (especially by scanning electron microscopy with energy-dispersive X-ray spectrometry, SEM/EDX) was used for the chemical, microphysical and optical aerosol characterization at Sede Boker in the Negev Desert, Israel. By making use of the comprehensive data set, estimations were made of the impact of the see breeze on the aerosol radiative effect and of the aerosol core-shell structure and the implication for remote sensing. This is clearly a thorough study. However, the results are mainly based on aerosol samples that were respectively collected before and during the sea breeze on 16 August 2012. One may wonder how representative the situation on that day was for the remainder of the summertime or whether it

may also occur for the other seasons. As indicated below, the manuscript has other shortcomings, so that some revision is needed before it can be published in ACP.

Specific comments:

1. A comprehensive SEM/EDX characterization for aerosol samples from the same Sede Boker site was previously performed by Sobanka et al. (J. Atmos. Chem., 44, 299-322, 2003). In this study coarse (2-10 $\mu$m aerodynamic diameter, AD) and fine (<2 $\mu$m AD) aerosol samples from summer and winter campaigns were analysed. Although the authors make reference to this paper in the Introduction, they fail to compare their particle classification presented in Figure 3 and their particle type proportions of Figure 8 with results from Sobanska et al. Some comparison with the summer data of Sobanka et al. is necessary.

2. Page 3, lines 18-20: A literature reference would be welcome for the statement in this sentence.

3. Page 6, lines 1: It unclear what it meant by "data correspond to the quality level 1.5". Some explanation is needed here.

4. Page 19, line 26, and page 31, Figure 8: The use of PM1 and PM2.5, as used here, is very confusing. These terms are normally used to denote particles smaller than 1 and 2.5 $\mu$m AD, respectively, whereas they clearly denote other size ranges in the current manuscript. I recommend replacing PM1 by PM2.5-1 and PM2.5 by PM10-2.5.

5. Technical and other (mostly minor) corrections:

- page 1, line 19: replace "found be" by "found to be".

- page 2, line 14: there is something grammatically wrong with "which hygroscopic".

- page 2, line 28: replace "site sometimes" by "site is sometimes".

- page 3, line 14: replace "program, e.g., (Ichoku" by "program (e.g., Ichoku".
- page 3, line 18: replace "area of" by "areas of".

- page 3, line 22: replace "show generally" by "showed generally".

- page 3, line 24: replace "Although, the" by "Although, the".

- page 3, line 32: I presume that "(4)" should be replaced by an appropriate literature reference.

- page 4, line 8: replace "Although, the" by "Although, the".

- page 4, line 25: replace "The Ångström" by "An Ångström".

- page 8, line 9: replace "by (Eilers, 2003; Eilers and Boelens, 2005)" by "by Eilers (2003) and Eilers and Boelens (2005)".

- page 9, line 25: replace "in details the" by "in detail the".

- page 10, line 34: replace "2 mm in contrast to 2.5 – 3 mm" by "2 $\mu$m in contrast to 2.5 – 3 $\mu$m".

- page 11, line 2: replace "in visible" by "in the visible".

- page 11, line 14: replace "that sensitivity" by "that the sensitivity".

- page 12, line 11: replace "fine fractions" by "fine fraction".

- page 12, line 27: replace "in (Reid et al., 2003)" by "in Reid et al. (2003)".

- page 12, line 27: replace "per particles type" by "per particle type".

- page 15, lines 10-11: replace "in (Derimian et al., 2016)" by "in Derimian et al. (2016)".

- page 15, line 26: replace "in (Derimian et al., 2016)" by "in Derimian et al. (2016)".

- page 16, lines 18 and 23: replace "Dubovik et al., (2000)" by "Dubovik et al. (2000)".

- page 16, line 29: replace "in (Dubovik et al., 2000)" by "in Dubovik et al. (2000)".

**ACPD**

- page 17, line 1: replace "in (Dubovik et al., 2000)" by "in Dubovik et al. (2000)".

- page 17, line 20: there should be space before the "are" in "are 440".

- page 17, line 22: replace "it also" by "it is also".

- page 18, line 5: replace "14b)," by "14 b,".

- page 18, line 16: replace "Also, notable" by "Also notable".

- page 18, line 34: replace "of (Dubovik et al., 2000)" by "of Dubovik et al. (2000)".

- page 19, line 7: replace "in details" by "in detail".

- page 20, lines 22-23: the quotation marks are unpaired.

- page 21, line 9: replace "J ATMOS OCEAN TECH" by "J. Atmos. Ocean. Tech.".

- page 22, line 18: the journal name should be abbreviated.

- page 24, line 3: the journal name should be abbreviated.

- page 24, lines 5-8: there are several problems with this reference.

- page 25: the heading of Table 1 should be above the table instead of below it; furthermore, replace "Relative humidity" by "relative humidity".

- page 27, line 5: there is something wrong with "arrival occurred on"; rephrasing is needed.

- page 33, within the top right panel of Figure 11: replace "C" by "Ca".

- page 34, line 7: I cannot see any colored arrows in the figure.

- page 36, line 5: it is unclear what "in this section" is doing here.

---

## Referee Comment (RC3) · Anonymous Referee #2 · 8 Mar 2017

This is an interesting and well written article that describes how the composition of aerosols at an inland site can change dramatically on a daily basis because of the influence of sea breezes. The authors point out that this can have a significant impact on the atmospheric radiative effect at the site. I have only a few suggestions for improvement.

Major issues

Page 11, lines 1-24 and Figure 7: Interesting discussion about how the refractive index changes with air mass and water vapor. The authors use standard deviations for the error bars in panels c-f of Figure 7 to understand the differences in the observations during low and high water vapor periods. However, it would be more useful to use the

standard deviation of the means for this application (i.e., SDOM, or standard errors). This will decrease the contribution of random noise to the size of the errorbars, and it will provide the reader with an understanding of whether these differences are statistically significant at the 1-sigma level (i.e., datasets with overlapping SDOM errorbars are not significantly different). The authors should also indicate how many data points are used to compute the means in panels c-f.

Page 11, lines 18-24: The authors bring up the topic of ssa in this paragraph, but don't really take it anywhere. You could isolate the effect of refractive index on the ssa for Aug 16 w/o much work though. . . that is, compute the ssa of the sea breeze aerosols using the SD of the pre-breeze particles. This will provide a Delta ssa associated with the size change. Similarly, you could compute the ssa of the pre-breeze particles using the refractive index of the sea-breeze particles; this will provide a Delta ssa associated with refractive index. This type of calculation can provide the reader an idea of how much of the ssa change is associated with size and how much is associated with composition, and it will make this paragraph more interesting.

Figures 8 & 9: Is there a discrepancy here?... The coarse mode is dominated by dust before the sea breeze in Figure 8, but Figure 9 indicates that there are more marine particles than dust particles at all radii > 0.5 um.

I really enjoyed the analysis of the effect of core-shell morphology on the AERONET retrievals (Section 7). I have a couple of additional points that I believe are worth including in the manuscript:

+ Water shell thicknesses of 10% and 40% correspond to geometric hygroscopic growth factors of 1.11 and 1.67 (GF = r / r_core). A value of GF = 1.11 seems reasonable, but GF=1.67 is a rather large value to obtain at ambient relative humidities (your figures indicate typical RHs of ∼60% for the sea breezes). These large growth factors are not impossible (especially since you are observing significant fractions of marine aerosols), but it would be worthwhile to discuss these GFs in the context of

TDMA measurements found in the literature. Swietlicki et. al. (Tellus 2008, 60B), for instance, provides a nice overview for measurements at 90% RH.

+ Level 2 AERONET retrievals do not include retrievals with residuals greater than 5-8% (depending upon AOT); thus, the 40% coating cases would not make it through the Level 2 AERONET screening, since the residual for that case is 14%. It is important to point this out to the reader, as it demonstrates that AERONET has the ability to omit cases where the aerosol morphology differs drastically from the morphology assumed in the retrieval. This is a much different conclusion than "the retrieval gets it wrong" for such cases.

Page 17, line 33: "The retrieved refractive indexes significantly exceed those of the core,..." This is somewhat unintuitive, so it would be worthwhile to explain why this happens in a sentence or two.

Minor issues

Page 4: Authors discuss the relationship between the Angstrom exponent and aerosol particle size, without citing the literature. They should provide one or more citations for uninitiated readers.

Page 10, line 24: "Figures 5c and d..." should refer to Figure 6. Page 10, line 34: "Also the maximum of the coarse mode..." should be "Also the maximum radius of the coarse mode..." Page 10, line 34: Replace millimeters (mm) with micrometers (um).

Figure 11, upper right panel: label should be Ca instead of C, right? Page 14 and Figure 12: There is much discussion about the colored arrows in Figure 12, but I do not see any arrows in my copy. Page 15, line 6: What is the wavelength range covered by the SolRad-Net pyranometer? Page 15, line 26: Figures 12c,d should be Figures 13c,d...

Page 16, line 2: The atmospheric radiative effect is related to the SSA, so you could tie this into your earlier discussion of SSA. That is, you could compute the radiative effect

using pre-breeze SD and sea breeze refractive indices to estimate the effect of size on the radiative effect (by comparing to the sea breeze computations that you have already done); likewise, computations utilizing the pre-breeze SD with both pre-breeze and sea-breeze refractive indices can be used to estimate the effect of composition on the radiative effect. I include this item as a "Minor Issue" because it would be a nice addition that will make the paper more interesting, but it is not something that is absolutely necessary for publication.

Page 17, line 27: "Note that the refractive index used in the case of homogeneous particles is the same as that of the core." I think that you should move this sentence to the end of the previous paragraph, as I was looking for this information earlier on.

---

## Author Comment (AC1) · 25 May 2017

**Author's response to Referee #1**

The thoughtful reading and the time dedicated by the reviewer are highly appreciated. The provided major as well as the minor comments are an important feedback that enabled better focusing of the scientific content and improvement of the manuscript quality. Below please find our point-to-point replies. The responses to the reviewer comments are given in blue text; the original reviewer comments are in black text.

**Anonymous Referee #1**

A combination of remote sensing observations, in-situ measurements and chemical analysis at the individual particle scale (especially by scanning electron microscopy with energy-dispersive X-ray spectrometry, SEM/EDX) was used for the chemical, microphysical and optical aerosol characterization at Sede Boker in the Negev Desert, Israel. By making use of the comprehensive data set, estimations were made of the impact of the see breeze on the aerosol radiative effect and of the aerosol core-shell structure and the implication for remote sensing. This is clearly a thorough study. However, the results are mainly based on aerosol samples that were respectively collected before and during the sea breeze on 16 August 2012. One may wonder how representative the situation on that day was for the remainder of the summertime or whether it may also occur for the other seasons. As indicated below, the manuscript has other shortcomings, so that some revision is needed before it can be published in ACP.

Regarding the question "…how representative the situation on that day was for the remainder of the summertime or whether it may also occur for the other seasons."
In general, the information that allows to conclude about the representativeness is provided in the initial version of the manuscript, but it was probably not emphasized and complete. For example, a time series of various observations in Fig. 4 shows occurrence of the sea breeze conditions on eight of nine presented days (including August 16). Note that these observations are repetitive. In addition, it is reported that the sea breeze is clearly observed in meteorological data on 51 days, in the period from June to August 2012, which is almost 60 % of the time. Also, it is reported that similar abrupt increases in the AOT can be observed in the AERONET data during summer time of all preceding and subsequent years.
Regarding the other seasons: the air mass transport from the Mediterranean Sea with high humidity and pollutants is a known characteristic for the summer period in the Negev desert (Andreae et al., 2002; Derimian et al., 2006; Karnieli et al., 2009; Maenhaut et al., 2014). As the effect on aerosol microphysics is mainly associated with the mixed and humid air mass transported from the Mediterranean Sea, it is expected that the summer season is the most affected.
We also would like to mention that the similar aerosol samplings (with and without sea breeze) were conducted on other days during the observation period. The compositional characteristics of particles were quite similar when all other optical and meteorological characteristics are occurring repeatedly. August 16[th] was selected for more detailed analysis because the sampling conditions, the selected timing for the sampling (i.e., start time, duration) were the most favorable on this day for discussing the variability of physical and chemical characteristics with respect to the variability of optical measurements.

Specific comments:

1. A comprehensive SEM/EDX characterization for aerosol samples from the same Sede Boker site was previously performed by Sobanska et al. (J. Atmos. Chem., 44, 299-322, 2003). In this study coarse (2-10 µm aerodynamic diameter, AD) and fine (<2 µm AD) aerosol samples from summer and winter campaigns were analysed. Although the authors make reference to this paper in the Introduction, they fail to compare their particle classification presented in Figure 3 and their particle type proportions of Figure 8 with results from Sobanska et al. Some comparison with the summer data of Sobanska et al. is necessary.

In line with previous works done on the same sampling site in the summer period, sea salt and mineral dust were reported to be the prevailing particle types (Sobanska et al., 2003; Formenti et al., 2001; Maenhaut et al., 1997). In our particle classification, we define one single "Dust" particle type by grouping same types of mineral dust (aluminosilicate, $CaCO_3$, $CaSO_4$, $SiO_2$, FeOx, $TiO_2$, mixed dust and transformed mineral dust) as those obtained by hierarchical cluster analysis (HCA) reported in Sobanska et al. In our study, fresh and aged sea salt particles are grouped in one single "Marine" particle type while in Sobanska et al. fresh and aged sea salt are divided into two types with aged sea salt particles always associated to the coarse fraction (2-10 µm in aerodynamic diameter) and aged sea salt associated to the fine fraction (<2 µm in aerodynamic diameter). Mg-, S-, K-rich particles were grouped in the "Other" type and correspond to the "S-only" and "Industrial mix sulphate and carbonaceous" in Sobanska et al. We did not find any Pb- or Zn-rich particles in our samples compared to Sobanska et al. The main difference in our particle classification lies in the type "Not classified" in Sobanska et al. that corresponds to our "Mixed Dust/Marine" type as we specifically focused our work on the sea breeze effect in a desert setting contrary to the emphasis on local dust events in Sobanska et al.

Furthermore, in Sobanska et al. the authors specified that on a specific day in summer time, a high proportion of sea salt (35% in the coarse size fraction PM2-10 and 12% in the fine fraction PM2) and mixed sea salt/mineral dust (~15% in the fine fraction PM2) were found and were representative for a marine source contribution. On the same day, the authors reported a high proportion of aluminosilicates (~30%) and $CaCO_3$ (~17%) in approx. the same proportion in fine and coarse fractions. Given that the sampling duration was from 08:52-19:30 local time, the chemical analysis of individual particles is representative of the average composition including before/during/after sea breeze. In our case, a short-term particle sampling started on the onset of the daily sea breeze enabled us to investigate specifically the characteristics of particle composition (start and end of daily sea breeze at 16:00 and about 18:00 local time, respectively). In our study, "Dust" and "Mixed Dust/Marine" particles account for 10-56% and 5-27%, respectively, depending on the size fraction and before/during sea breeze. Besides the fact that fine and coarse fractions are not exactly equivalent in the two studies, on overall, our results are consistent with Sobanska et al.
This point has been added in the manuscript by the following sentences inserted in page 12, line 14: "These results are consistent with those obtained by Sobanska et al. (2003) at the same sampling site on a specific day in the summer period (sampling duration includes before/during/after sea breeze): a high proportion of sea salt (35% in the coarse size fraction PM2-10 and 12% in the fine fraction PM2) and mixed sea salt/mineral dust (~15% in the fine

fraction PM2) representative of a marine source contribution. In addition, they reported a high proportion of aluminosilicates (~30%) and CaCO$_3$ (~17%) in approximately the same proportion in fine and coarse fractions."

2. Page 3, lines 18-20: A literature reference would be welcome for the statement in this sentence.
It is (Dayan and Rodnizki, 1999) that is cited in the next sentence. The phrase is modified to make it clear.

3. Page 6, lines 1: It unclear what it meant by "data correspond to the quality level 1.5". Some explanation is needed here.
Corrected. The explanation is "The data correspond to the quality level 1.5, which means that the data have been cloud screened and cleared of any operational problems."

4. Page 19, line 26, and page 31, Figure 8: The use of PM1 and PM2.5, as used here, is very confusing. These terms are normally used to denote particles smaller than 1 and 2.5 µm AD, respectively, whereas they clearly denote other size ranges in the current manuscript. I recommend replacing PM1 by PM2.5-1 and PM2.5 by PM10-2.5.
Absolutely agree. It is corrected in the revised version.

5. Technical and other (mostly minor) corrections: - page 1, line 19: replace "found be" by "found to be". - page 2, line 14: there is something grammatically wrong with "which hygroscopic". - page 2, line 28: replace "site sometimes" by "site is sometimes".  - page 3, line 14: replace "program, e.g., (Ichoku" by "program (e.g., Ichoku".
- page 3, line 18: replace "area of" by "areas of".
- page 3, line 22: replace "show generally" by "showed generally".
- page 3, line 24: replace "Although, the" by "Although, the".
- page 3, line 32: I presume that "(4)" should be replaced by an appropriate literature reference.
- page 4, line 8: replace "Although, the" by "Although, the".  - page 4, line 25: replace "The Ångström" by "An Ångström".
- page 8, line 9: replace "by (Eilers, 2003; Eilers and Boelens, 2005)" by "by Eilers (2003) and Eilers and Boelens (2005)".
- page 9, line 25: replace "in details the" by "in detail the".  - page 10, line 34: replace "2 mm in contrast to 2.5 – 3 mm" by "2 µm in contrast to 2.5 – 3 µm".  - page 11, line 2: replace "in visible" by "in the visible". - page 11, line 14: replace "that sensitivity" by "that the sensitivity".  - page 12, line 11: replace "fine fractions" by "fine fraction".  - page 12, line 27: replace "in (Reid et al., 2003)" by "in Reid et al. (2003)".  - page 12, line 27: replace "per particles type" by "per particle type". - page 15, lines 10-11: replace "in (Derimian et al., 2016)" by "in Derimian et al. (2016)". - page 15, line 26: replace "in (Derimian et al., 2016)" by "in Derimian et al. (2016).  - page 16, lines 18 and 23: replace "Dubovik et al., (2000)" by "Dubovik et al. (2000)". - page 16, line 29: replace "in (Dubovik et al., 2000)" by "in Dubovik et al. (2000)".
- page 17, line 1: replace "in (Dubovik et al., 2000)" by "in Dubovik et al. (2000)". - page 17, line 20: there should be space before the "are" in "are 440".  - page 17, line 22: replace "it also" by "it is also".  - page 18, line 5: replace "14b)," by "14 b,".

- page 18, line 16: replace "Also, notable" by "Also notable".
- page 18, line 34: replace "of (Dubovik et al., 2000)" by "of Dubovik et al. (2000)".
- page 19, line 7: replace "in details" by "in detail".
- page 20, lines 22-23: the quotation marks are unpaired.
- page 21, line 9: replace "J ATMOS OCEAN TECH" by "J. Atmos. Ocean. Tech.".
- page 22, line 18: the journal name should be abbreviated.
- page 24, line 3: the journal name should be abbreviated.
- page 24, lines 5-8: there are several problems with this reference.
- page 25: the heading of Table 1 should be above the table instead of below it; furthermore, replace "Relative humidity" by "relative humidity".
- page 27, line 5: there is something wrong with "arrival occurred on"; rephrasing is needed.
- page 33, within the top right panel of Figure 11: replace "C" by "Ca". - page 34, line 7: I cannot see any colored arrows in the figure. - page 36, line 5: it is unclear what "in this section" is doing here.

Thank you very much for taking time and providing all these technical, but essential correction. All these corrections are considered in the revised version.

---

## Author Comment (AC2) · 25 May 2017

**Author's response to Referee #2**

The thoughtful reading and the time dedicated by the reviewer are highly appreciated. The provided major as well as the minor comments are an important feedback that enabled better focusing of the scientific content and improvement of the manuscript quality. Below please find our point-to-point replies. The responses to the reviewer comments are given in blue text; the original reviewer comments are in black text.

**Anonymous Referee #2**

This is an interesting and well written article that describes how the composition of aerosols at an inland site can change dramatically on a daily basis because of the influence of sea breezes. The authors point out that this can have a significant impact on the atmospheric radiative effect at the site. I have only a few suggestions for improvement.
Major issues
Page 11, lines 1-24 and Figure 7: Interesting discussion about how the refractive index changes with air mass and water vapor. The authors use standard deviations for the error bars in panels c-f of Figure 7 to understand the differences in the observations during low and high water vapor periods. However, it would be more useful to use the standard deviation of the means for this application (i.e., SDOM, or standard errors). This will decrease the contribution of random noise to the size of the errorbars, and it will provide the reader with an understanding of whether these differences are statistically significant at the 1-sigma level (i.e., datasets with overlapping SDOM errorbars are not significantly different). The authors should also indicate how many data points are used to compute the means in panels c-f.

Thank you for this suggestion, the standard errors are presented in the revised version. It is indeed appropriate. The cases where the standard errors overlap the means are indicated in the text as non-significant (imaginary refractive indexes during dust season, Fig. 7f). The number of data points used is added as well.

Page 11, lines 18-24: The authors bring up the topic of ssa in this paragraph, but don't really take it anywhere. You could isolate the effect of refractive index on the ssa for Aug 16 w/o much work though. . . that is, compute the ssa of the sea breeze aerosols using the SD of the pre-breeze particles. This will provide a Delta ssa associated with the size change. Similarly, you could compute the ssa of the pre-breeze particles using the refractive index of the sea-breeze particles; this will provide a Delta ssa associated with refractive index. This type of calculation can provide the reader an idea of how much of the ssa change is associated with size and how much is associated with composition, and it will make this paragraph more interesting.

Absolutely agree that the suggested calculation makes the analysis more interesting. The Delta SSAs due to the size changes and the compositional changes are added, and the next discussion is provided: "Because both the size distribution and the complex refractive index change during the sea breeze, it is interesting to evaluate their specific contribution to the changes in the SSA. To address this, we calculate the SSA assuming that only the size distribution is changing while the refractive index is the same and vice versa. The difference

in the SSA of the before-sea-breeze aerosol model minus the SSA of the modified aerosol model is -0.002/-0.001/0.001/0.003 for the size change and 0.015/0.009/0.003/0.003 for the refractive index change. The calculated differences show that the scattering effectiveness increases at the shorter and decreases at the longer wavelengths due to the size change, and decreases at all the wavelengths due to the compositional change. It shows that there is a partial compensation of the decrease in SSA at the shorter wavelengths because of the size shift."

It has also be mentioned that a mistake in the reported SSA values was found during the revision. However, it does not change the reported tendencies, conclusions or any other reported results.

Figures 8 & 9: Is there a discrepancy here?... The coarse mode is dominated by dust before the sea breeze in Figure 8, but Figure 9 indicates that there are more marine particles than dust particles at all radii > 0.5 um.

It should be clarified in the text that Figures 8 & 9 are not directly comparable. In fact, one must keep in mind the differences in measurement techniques (aerodynamic diameter for cascade impactors in Figure 8 versus geometric diameter determined by electron microscopy in Figure 9), that is:
- Figure 8 shows the relative proportions of particle types as a function of the aerodynamic size range. The size-segregated sampling by cascade impaction is based on an aerodynamic cut-off diameter with 50% efficiency $D_{ae,50}$ depending on particle density. The Dekati impactor used in this study is calibrated for a particle density of 0.93 g cm$^{-3}$ (Marjamaki et al., 2000). This last information is now added in the manuscript (page 6, line 30-31 of the initially submitted version). As can be read on page 11, line 31, the "coarse fraction" refers to particles collected on the stage with the aerodynamic diameter range 2.5-10 µm and the "fine fraction" refers to particles collected on the stage with the aerodynamic diameter range 1-2.5 µm.
- On the other hand, the number size distribution presented in Figure 9 reports radii values for all analyzed particles collected on both stages (1-2.5 µm and 2.5-10 µm). The geometric radius in this analysis is derived from equivalent circle area of the 2D-projected particle on SEM images.
The next clarification is added in the text of the revised version (at the beginning of section 5.2): "In addition, it should be realized that the size distributions of the particle types in Fig. 9 are not directly comparable to the relative proportions of particle types per size fraction in Fig. 8. This is because the particle type proportions reported in Fig. 8 are for the size fractions of a cascade impactor, which are defined by an aerodynamic cut-off diameters, while Fig. 9 presents the geometric radius derived from equivalent circle area of particles observed by SEM."

I really enjoyed the analysis of the effect of core-shell morphology on the AERONET retrievals (Section 7). I have a couple of additional points that I believe are worth including in the manuscript:
+ Water shell thicknesses of 10% and 40% correspond to geometric hygroscopic growth factors of 1.11 and 1.67 (GF = r / r_core). A value of GF = 1.11 seems reasonable, but GF=1.67 is a rather large value to obtain at ambient relative humidities (your figures indicate typical RHs of 60% for the sea breezes). These large growth factors are not impossible (especially since you are observing significant fractions of marine aerosols), but it would be worthwhile

to discuss these GFs in the context of TDMA measurements found in the literature. Swietlicki et. al. (Tellus 2008, 60B), for instance, provides a nice overview for measurements at 90% RH.

Thank you, it is indeed important to link between the geometric hygroscopic growth factor and thickness of shell that is used in the simulations. The next discussion is added on page 17, after line 2 (initial version):
"Three simplified scenarios are considered: first – the particles are homogeneous, second and third – a liquid water layer coats the particles with a thickness that corresponds to 10 % and 40 % of the total particle radius, respectively. This percentage is assumed because at a thickness of about 10 % the differences in optical characteristics become notable and for about 40 – 50 %, the residual of the fit in the inversion procedure reaches a maximum. This indicates the largest discrepancy between the core-shell model and the particle homogeneity assumption as used in the inversion. To put the percentages used here in the context of real observations, it can be represented in terms of the widely used geometric hygroscopic growth factor, which is the ratio between humidified and dry particle diameter. Thus, 10 % corresponds to a growth factor of 1.11, which can be defined as a low to moderate value, and 40 % corresponds to 1.67, which is near the upper limit of values in the review by Swietlicki et al., (2008), for instance. It is noteworthy that our tests show important differences in optical characteristics and increased residuals of fit also for 30 and 20 % shell thickness. In fact, the effect of the coating also depends on the shape of the particle size distribution and the contrast in refractive indexes of core and shell, therefore, the subject merits some more detailed studies."

+ Level 2 AERONET retrievals do not include retrievals with residuals greater than 5- 8% (depending upon AOT); thus, the 40% coating cases would not make it through the Level 2 AERONET screening, since the residual for that case is 14%. It is important to point this out to the reader, as it demonstrates that AERONET has the ability to omit cases where the aerosol morphology differs drastically from the morphology assumed in the retrieval. This is a much different conclusion than "the retrieval gets it wrong" for such cases.

I appreciate sharing of this thought, it is included in the related section of the manuscript. The phrase in bold is included (p.18, line 21, initial version): "The residual of the fit is quite high, which means that a physical interpretation of the retrieved microphysical parameters should be done with caution. **In addition, retrievals with high residuals are generally screened in final products and therefore cases where the aerosol morphology differs drastically from the morphology assumed in the retrieval algorithms may be omitted.** However, the obtained high fit error show the sensitivity of the measurements to the core-shell structure."

Page 17, line 33: "The retrieved refractive indexes significantly exceed those of the core,..." This is somewhat unintuitive, so it would be worthwhile to explain why this happens in a sentence or two.
The next sentences are added: "It is expected that in the case of mixed aerosol the values of the retrieved refractive index will be in between the refractive indexes of the two components. The fact that the retrieved values are greater and that the size distribution is modified suggests that the inversion algorithm attempts to compensate the specific particle morphology by an exceptional aerosol model."

Minor issues

Page 4: Authors discuss the relationship between the Angstrom exponent and aerosol particle size, without citing the literature. They should provide one or more citations for uninitiated readers.

The citations are added.

Page 10, line 24: "Figures 5c and d..." should refer to Figure 6.

Corrected.

Page 10, line 34: "Also the maximum of the coarse mode..." should be "Also the maximum radius of the coarse mode..."

Corrected.

Page 10, line 34: Replace millimeters (mm) with micrometers (um).

Corrected.

Figure 11, upper right panel: label should be Ca instead of C, right?

Corrected.

Page 14 and Figure 12: There is much discussion about the colored arrows in Figure 12, but I do not see any arrows in my copy.

Corrected. Sorry for this moment of distraction during the image conversion.

Page 15, line 6: What is the wavelength range covered by the SolRad-Net pyranometer?

The wavelength range is 0.3 – 2.8 μm. The information was provided on page 5, line 30. However, the information that is missing is the wavelength range of the calculated solar flux (0.2 – 4 μm). The discrepancy between the spectral range of measurements and calculations should be mentioned because it leads to about 3 % bias. At the same time, this discrepancy is important only for an inter-comparison of the measured and the calculated flux (note that 3 – 5 % is a usual accuracy of the flux measurements), but not for the presented analysis of relative perturbation. It is because the perturbations in the measured and the calculated solar flux are estimated separately. The sentences in bold are added on page 15, lines 9-11: "To evaluate the sea breeze induced radiative effect, we calculated the solar fluxes and the net instantaneous direct aerosol radiative effect using a computational tool described in (Derimian et al., 2016). **Note that the calculated solar flux is for the wavelength range of 0.2 – 4.0 μm, while the measured is for 0.3 – 2.8 μm, which implies about 3 % bias due to the cut-off of the spectral range (note that the accuracy of the measurements themselves is about 3 – 5 % as well). Nevertheless, this discrepancy in the spectral ranges does not affect analysis of the relative perturbation of the solar flux when evaluated using the measurements or the calculations separately.**"

Page 15, line 26: Figures 12c,d should be Figures 13c,d...

Corrected.

Page 16, line 2: The atmospheric radiative effect is related to the SSA, so you could tie this into your earlier discussion of SSA. That is, you could compute the radiative effect using pre-breeze SD and sea breeze refractive indices to estimate the effect of size on the radiative

effect (by comparing to the sea breeze computations that you have already done); likewise, computations utilizing the pre-breeze SD with both pre-breeze and sea-breeze refractive indices can be used to estimate the effect of composition on the radiative effect. I include this item as a "Minor Issue" because it would be a nice addition that will make the paper more interesting, but it is not something that is absolutely necessary for publication.

We thank the Reviewer for this suggestion. This indeed could be an interesting addition. However, there is a problem to distinguish between the effect of the changing microphysics/composition and the changing AOT on the radiative effect. That is, not only the shape of the size distribution will change, but also aerosol volume concentration, which means a change of AOT. In order to isolate the effect of microphysics, the radiative efficiency should be used (radiative effect normalized by AOT). However, a nonlinear dependence exists between aerosol radiative effect and AOT. Even the small effect of this nonlinearity can be comparable with the fine effect of difference in aerosol microphysics. All this makes this type of analysis quite delicate. We therefore, prefer to avoid this complex discussion.

Page 17, line 27: "Note that the refractive index used in the case of homogeneous particles is the same as that of the core." I think that you should move this sentence to the end of the previous paragraph, as I was looking for this information earlier on.

It is moved up.

---

## Author Comment (AC3) · 25 May 2017

**Author's response to Referee #3**

The thoughtful reading and the time dedicated by the reviewer are highly appreciated. The provided major as well as the minor comments are an important feedback that enabled better focusing of the scientific content and improvement of the manuscript quality. Below please find our point-to-point replies. The responses to the reviewer comments are given in blue text; the original reviewer comments are in black text.

**Anonymous Referee #3**

Review of 'Effect of sea breeze circulation on aerosol mixing state and radiative prop- erties in a desert setting' by Derimian et al.
The paper studies the modifications in summertime sea breeze conditions of the aerosol compositional, microphysical, optical, and radiative properties at an inland lo- cation of the Negev Desert (Israel). It is well written and easy to read. The study is original and the scientific methodology sound. The complementarity of the inversion of remote sensing (sunphotometer) observations and of the direct, and I suppose time-consuming, off-line individual analysis for characterizing the particles is particularly interesting. The authors evidence for the first time the significant influence of the daytime intrusions of marine air on the aerosol characteristics at such a remote place.
Not only the composition, but also to internal structure of the particles greatly differ in the pre- and post- see breeze situations. In particular, the proportion of mineral (desert-dust) particles surrounded by a coating is unexpectedly large in both cases, which contradicts the common assumption that desert dust is hydrophobic. These modifications of the aerosols characteristics need to be taken into account for quantifying their radiative effect. Finally, the numerical simulations made by the authors show that the current remote sensing inversion algorithms need to be modified in order to take the core-shell structure of the particles into account. My opinion is that this paper deserves publication provided the following concerns are addressed: General comment: After reading the paper, one is left with the impression (see for instance lines 21-23, page 2) that the aerosols initially present at Sede Boker are modified by the arrival of the sea breeze. In fact, these pre-existing aerosols are most probably blown further downwind of the experimental site by the breeze and replaced by new freshly advected particles. The authors make an exhaustive and quite interesting comparison of the characteristics of these two sets of particles, but if the particles are not the same, is it possible to conclude that the size increase observed after the arrival of the sea breeze could be due to the water vapor uptake? More generally, the mineral particles observed during the marine intrusions probably have a long history of coexistence with the other species (sea-salt and anthropogenic aerosols and gases), what's more in humid air-masses. Therefore, they are more liable to have formed internal mixtures than the resident aerosol of Sede Boker.

Agree and reworked the text in order to emphasize that the new air masses arrive and replace the previously existing aerosol. Regarding the question: "…if the particles are not the same, is it possible to conclude that the size increase observed after the arrival of the sea breeze could be due to the water vapor uptake?"

Thanks for this question, indeed, the conclusion should be better focused. It should be emphasized that the water uptake is a hypothesis. This hypothesis is suggested as an explanation of the aerosol size shift based on the findings that: 1) the particles are more hygroscopic and 2) the air contains more water. Thus, the water uptake is suggested as the most probable reason for the aerosol size increase, relative to the background conditions. Of course, other reasons for the size increase are also possible, for example, it can be freshly advected dust. However, the freshly advected large particles are expected to settle quite fast or mostly be present near the surface. The photometric measurements, however, are for the total atmospheric column and are thus not expected to be significantly influenced by the near-surface conditions. In order to support this supposition, we conducted a supplementary analysis by sorting the retrieved size distributions (data for summer 2012) by near the ground measured wind speed and RH. The figure below shows normalized volume size distributions. It shows that the shape of the distribution does not change significantly for low versus high wind speed. However, it does change when the RH is changing; the tendency to the shift of the fine mode is consistent with the size distribution during the sea breeze on August 16[th].

[Figure]

It is unlikely that the size shift is due to freshly emitted from the surface dust also because the main increase in the particles size is in the fine mode fraction, which is dominated by mixed dust/marine and pollutants and not by pure dust. The fine mode is more strongly influenced by pollutants and mixed dust/marine aerosol which are highly hygroscopic. It does not, however, diminish the role of the apparent hygroscopicity of dust once reacted with marine particles or pollutants.

In summary, following the reviewer's comment, it is realized that the related discussions and conclusions should be focused better. The text in the new version of the paper is reworked accordingly.

Miscellaneous:

1) P. 7: What can the origin of the Ti-rich particles be?

Ti-rich particles presumably consist of $TiO_2$. The main source of $TiO_2$ particles suspended in the atmosphere is most likely from windblown mineral dust (e.g., Chen, H., Nanayakkara, C. E., & Grassian, V. H., Titanium dioxide photocatalysis in atmospheric chemistry. Chemical Reviews, 112(11), 5919-5948, 2012). In our case, Ti was often found mixed with other metallic elements including Fe, Ca, Si and Al that are typical constituents of windblown eroded soils.

On overall, Ti-rich particles accounted for 1.4% of all analyzed particles and occurred in both fine and coarse fractions.

2) P. 8, line 25; on Fig. 4a, the Angström exponent increases with the arrival of the sea breeze on 14 August. Is there a plausible explanation for this exception to the rule?

The Angström exponent is indeed increasing, but only before the sea breeze, at the moment of the sea breeze arrival, it rapidly decreases as in all other cases.

3) P. 10, line 8: the nephelometer 'dries'. . .

Corrected.

4) P.10, lines 12-13: couldn't the 'abrupt response' also be due to the increase of the aerosols concentrations and to a shift in their size?

Yes, the response in optical characteristics is indeed due to the changes in both, concentration and microphysics. The text is clarified as follows: "In summary, all the above mentioned observations of the aerosol optical properties in the solar spectrum and radiation in the thermal infrared wavelength region manifest a coherent abrupt response associated with the sea breeze arrival. An abrupt response in the aerosol optical characteristics can be due to a higher aerosol concentration, but also a change in the aerosol microphysical characteristics and influence of the increasing atmospheric water content. In order to examine the possible change in the aerosol microphysical parameters that take place during the sea breeze, we use the remote sensing observations …"

5) P.10, line 24: This is Fig. 6 (not 5)

Corrected.

6) P.10, line 33: The unit is µm not mm

Corrected.

7) Fig 6c and d : the blue line corresponds to WV larger than

The notation is correct there.

8) P.11; line 5: As said in the following sentences, the large standard deviation does not allow concluding that there is 'a decrease' of the mean real refractive index. I would remove the 'Curiously a decrease. . .'.

Please note that, as suggested by Reviewer #2, the standard deviations are replaced by standard errors, which are more appropriate to the discussion about significance of the measurements. The standard errors do not overlap the mean in the case of the real part. The corresponding text was rewritten.

9) P.11; line 31 and the notations of Fig. 8: Usually, PM1 and PM2.5 correspond to particles with diameters smaller than 1 and 2.5µm, respectively. Here, PM1 corresponds to particles with diameters between 1 and 2.5, and PM2.5 to the range 2.5-10µm. This is confusing.

Agree, the notation in Fig. 8 should correspond to the aerodynamic cut-off diameter of the impactor and not to the conventional PM. It is corrected to PM1-2.5 and PM2.5-10 in Fig. 8 and corresponds now to the explanations in the text, a clarification is also added in the figure caption.

10) P. 12, lines 15-16: the 'other' particles represent 7% of the coarse fraction but are said to be smaller than 1µm in diameter. Isn't this contradictory?

There is no contradiction in the data, but a confusion due to a difference in the measurements techniques. These particles were analyzed by SEM/EDX and found on the PM2.5-10 stage of the impactor. Because 2.5 µm is a cut-off diameter at 50 % of collection efficiency, particles smaller than 2.5 µm can also be present on this stage (the size-segregated sampling by cascade impaction is based on an aerodynamic cut-off diameter and depends on particle density). The diameter obtained by SEM observation is derived from particle's 2D-projection and corresponds to the geometric diameter of the equivalent circle area. Depending on the density, the aqueous content and morphology of ambient particles, it is possible that the SEM technique will provide geometric diameter smaller or bigger than the aerodynamic cut-off diameter of the impactor. A clarification is added in the section about aerosol sampling methodology (3.3.2).

11) P. 13, line 1: the authors say that the shift towards larger sizes of the marine particles during the sea breeze could be due to hygroscopic growth. Aren't the SEM observations made under a vacuum, i.e. in dry conditions? Moreover, if we go back to my first comment, please consider that the marine particles observed at the inland site before the arrival of the sea breeze might be more aged than the new ones. Consequently, their size-distribution might have been modified by the size-selective dry deposition process. For instance, I cannot help observing on Fig 9 that the very fine and the coarse particles present in the fresh marine air-masses (Fig. 9 b) have disappeared on Fig 9a, and that on the latter figure, only the particles with a diameter corresponding to the smallest deposition velocity (around 1µm) subsist.

Cascade impaction is conducted at ambient temperature and RH, and a dry size distribution is derived following the SEM observation. Indeed, the SEM observations were performed under high vacuum and water-solvated ions dehydrate in the SEM chamber, but due to the wettability of the substrate, initially hydrated sea salt particles appear generally in SEM pictures as flat particles (larger than thick) and often as rounded shape for aged marine particles, consisting of a core and a shell formed by residues as shown in Figure 10d. Thus, the geometric size of marine particles given by SEM may be slightly underestimated due to the low thickness of the border of the dehydrated particle but not as much as if marine particles were dehydrated before sampling. That is why the shift towards larger sizes of the marine particles during the sea breeze could be due to hygroscopic growth.

However, we also agree with the referee on the possibility of a shift towards larger marine particles of the coarse mode during sea breeze due to size-selective dry deposition processes that can occur during the aerosol transport. Thus, the text was modified considering this hypothesis (page 13, line 1): "This can be due to hygroscopic growth caused by the higher RH during the sea breeze, but size-selective dry deposition processes (Seinfeld and Pandis, 1998) can also take place during the aerosol transport since the wind speed and atmospheric residence time of marine particles are different before and during the sea breeze."

12) P. 13, line 24: There is no Mg in the composition of the calcite.

Mg is referring to dolomite. The text is modified. Both dolomite and calcite are minerals containing calcium carbonate. Dolomite differs from calcite because of the presence of magnesium (i.e., calcite ($CaCO_3$) mainly contains calcium carbonate and dolomite $CaMg(CO_3)_2$ contains calcium magnesium carbonate).

13) Fig. 11: On the upper right-hand panel, this should be Ca (not C).
Corrected.

14) Fig. 12: I cannot see the arrows mentioned on page 14.
Corrected. Sorry for this lack of attention during the image conversion.

15) P. 15, line 19: In the reduction of 5%, what are the respective shares of the 1) aerosol changes and 2) WV increase?
Thank you for this question. It is indeed important to distinguish between the effect of water vapor and aerosol. Supplementary calculations were conducted and the next discussion is added to the manuscript:
"… This amounts to 4.6 % reduction of the total solar flux that would reach the surface without the sea breeze effect. It should be realized, however, that the reduction in the solar flux is not only due to the change in aerosol properties, but also due to the increase in the water vapor content. In order to estimate the role of each component, additional calculations were conducted assuming that only the increase in the water vapor takes place, and then, assuming that only the aerosol properties change. The results show that the increase in the water vapor (from 1.62 to 2.13 g cm$^{-2}$) is responsible for a loss of 7.5 W m$^{-2}$ in the solar flux reaching the surface, while the change in the aerosol properties is responsible for 15.5 W m$^{-2}$ of the total 23 W m$^{-2}$ difference, which amounts to 1.5 and 3.1 %, respectively."

16) P. 15, line 26: this should be Fig 13, not 12.
Corrected.

17) Section 7: please, consider reformulating the whole section. It is much harder to follow than the rest of the paper. For instance, the reader discovers only on page 17 that forward calculations have been made (and with which inputs), then that different scenarios have been considered for inversion simulations.
Thank you for this feedback, the section was reorganized and some parts reformulated.

18) P.19, lines 2-5: could you be more specific regarding implications for the satellite and LIDAR inversions?
The sentences were modified as follows: "…we can also conclude that including backward scattering angles and polarimetric measurements present more sensitivity to the core-shell structure. This is because the main differences, due to the aerosol core-shell structure, are observed in the angular and polarimetric characteristics of the backward scattered light. Thus, since the backward scattering is a primary signal measured by satellites and LIDAR, important implications for these types of measurements are possible. For example, the aerosol core-shell structure will affect the lidar ratio and a parameterized core-shell aerosol model can be considered in satellites retrievals."